# Dynamical-VAE-based Hindsight to Learn the Causal Dynamics of Factored-POMDPs

## Abstract

Learning the underlying Markovian dynamics of an environment, from partial observations, is a key first step towards model-based reinforcement learning. Considering the environment as a Partially Observable Markov Decision Process (POMDP), state representations are typically inferred from the history of past observations and actions. Instead, we design a Dynamical Variational Auto-Encoder (DVAE) to learn causal Markovian dynamics from offline trajectories in a factored-POMDP setting. In doing so, we derive that incorporating future information is essential to accurately capture causal dynamics and the underlying Markovian states. Our method employs an extended hindsight framework that integrates past, current, and multi-step future information, to infer hidden factors in a principled way, while simultaneously learning transition dynamics as a structural causal model. Our framework is derived from maximizing the log-likelihood of complete trajectories factorized in time and state. Empirical results in a 1-hidden factored-POMDP setting, reveal that this approach uncovers the hidden factor up to a simple transformation, as well as the transition model and causal graph, more effectively than history based, typical 1-step hindsight based, and full trajectory bidirectional-RNN-based models.

## 1 Introduction

Accurately learning the underlying dynamics of an environment is essential for developing models that can reliably predict future states, particularly in partially observable settings (Wang et al., 2019; Moerland et al., 2023). Existing self-predictive approaches to state representation aim to learn a Markovian transition model (Ni et al., 2024). However, in partially observable contexts, the true underlying state remains hidden, making it necessary to construct an approximate belief state from prior state-action histories as a proxy for the latent state. This approach effectively reformulates the Partially Observable Markov Decision Process (POMDP) as a Markov Decision Process (MDP) that depends solely on past observations and actions to approximate the full state information (Åström, 1965; Subramanian et al., 2022). Such an approach may, in general, only lead to an approximation of the true underlying / generating MDP.

In online settings, the agent is limited to past information alone, but in offline RL or model learning, both past and future data around each time step are accessible. This availability raises the question of whether combining both past and future information can improve our ability to identify the generating MDP. By maximizing the log-likelihood of complete trajectories of observations and actions, we leverage the formalism of Dynamical Variational Auto-Encoders (DVAE) (Girin et al., 2020) to determine which elements of the past and future are essential for decoding unobservable variables at each time step.

We consider a factored-POMDP setting (Oliehoek et al., 2021), where the underlying MDP state is composed of multiple independent factors, some of which are observable while others remain hidden. This setting renders the environment partially observable while maintaining a low-dimensional state representation. We separate each unobservable state variable or factor into a deterministic hidden variable, and an exogenous stochastic one using the Reparameterization Lemma (Buesing et al., 2018). We derive that the 1-step past (including bootstrapped hidden), present, and future observables and actions are needed to identify deterministic unobserved hidden variables. We term our approach "DVAE-based hindsight" to contrast

it with prior hindsight methods for latent identification that utilized only the present and 1-step future information(Jarrett et al., 2023).

We utilize Causal Dynamical Learning (CDL) (Wang et al., 2022), employing Conditional Mutual Information (CMI), to learn a causal transition graph of the factored-MDP environment. The stationary Markovian transition model can be represented as a Directed Acyclic Graph (DAG), mapping the factored states and action at time step $t$ to the factored states at $t+1$. We extend CDL to a partially observable setting by learning to identify deterministic hidden variables and constructing the causal transition graph, combining the DVAE and CDL approaches in an end-to-end framework. We experimentally demonstrate the effectiveness of our approach compared to the history-based method(Littman & Sutton, 2001; Baisero & Amato, 2020; Ni et al., 2024), the earlier hindsight-based method(Jarrett et al., 2023) and a full trajectory bidirectional-RNN-based method, in a factored-POMDP setting (Oliehoek et al., 2021) with 1-hidden factor, as proof of principle on the advantages of our method.

## 2 Preliminaries and Problem Formulation

### 2.1 Partially Observable Markov Decision Processes (POMDPs)

A Markov Decision Process (MDP) in the context of reinforcement learning is defined by a tuple $(S, A, T_a, R_a)$, where $S$ is the set of states, $A$ the set of actions, $T_a(s'|s)$ the probability of transitioning from state $s$ to $s'$ under action $a$, and $R_a(s', s)$ the reward received for this transition. However, many real-world systems or environments are only partially observable. It is typically assumed that there exists an underlying or generating MDP that gives rise to a Partially Observable Markov Decision Process (POMDP) $(S, A, T_a, R_a, \Omega, O)$, where the states are not directly observable. Instead, we observe elements $o$ from a set $\Omega$, governed by conditional probabilities $O(o|s)$. A POMDP can be converted into an MDP (though this may not be the generating MDP), by relying solely on the history of observations and actions (Åström, 1965). This approach forms the basis for using a sequence of past observations (or their representation) and actions as a proxy, or belief state, for the environment's current state (Subramanian et al., 2022).

### 2.2 Problem formulation: Learning the causal dynamics underlying a factored-POMDP

Our objective is to learn the underlying state transitions and associated causal graph (represented in Figure 1) from offline data in a factored-POMDP environment.

In a factored-POMDP, the state $s$ is represented as a concatenation of observed and hidden state factors, denoted by $s = (o, h)$. The state transition probability distribution $T$ can be factorised as $T(s_{t+1}|s_t, a_t) = \prod_{j=1}^{d_S} p(s_{t+1}^j|s_t, a_t)$. Consequently, our goal reduces to learning the factored transitions $p(o_{t+1}^j|\{s_t^i\}_{i=1}^{d_S}, a_t)$ for $j = 1, \ldots, d_O$ and $p(h_{t+1}^j|\{s_t^i\}_{i=1}^{d_S}, a_t)$ for $j = 1, \ldots, d_H$. This enables constructing a causal graph between observed and hidden state factors (and actions) at each time step represented as nodes, with edges only from one time step to the next and not within a time step (See Figure 1).

However, in a general POMDP, the transition or observation distributions may not be factorizable. In such cases, representation learning becomes necessary to disentangle the observation vector into underlying state factors that are conditionally independent given their parents. A factored-POMDP (Oliehoek et al., 2021) allows us to focus on learning the underlying transition function and graph structure, without the added complexities of representation learning. A motivating example of such an environment is a medical scenario involving time series of patient records that measure only some factors. For the applicability of our framework, these factors must be independent given the values of their parent factors at the previous time step. The objective is to uncover other latent hidden factors necessary for constructing an underlying Markovian transition model of both observable and hidden factors (the full state) and predicting disease progression (reward), as in ICU datasets (Komorowski et al., 2018).

**Definition 1 (Factored-POMDP).** A factored partially observable Markov decision process (Oliehoek et al., 2021) is defined as a tuple $\langle S, O, H, A, T, R, \bar{O} \rangle$ where:

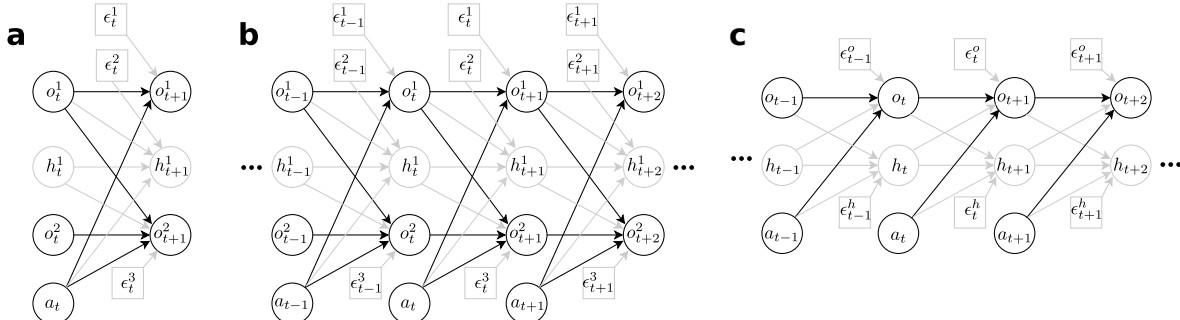

Figure 1: **(a)** The stationary transition model of a factored-POMDP is shown as a Structural Causal Model (SCM) from time step $t$ to $t+1$. The factored states are represented as circle nodes, which are deterministic as per Eq. 1. They can be either observed (black) or hidden (gray). Gray squares represent unobserved exogenous (i.e. no parents) stochastic nodes. The arrows connecting nodes represent directed causal edges from parents to children. The connectivity of the deterministic nodes is only an example. **(b)** The stationary transition model can be unrolled over time, by repeating the graph in panel **(a)** over multiple time steps, to obtain a SCM for a full trajectory. **(c)** We collect hidden factored states into vector $h$, and observable factored states into vector $o$ while maintaining the underlying causal model. In general, if actions are selected based on states rather than randomly, the action nodes in the unrolled transition graphs (**(b)** and **(c)**) would explicitly depend on states at the previous time step, whereas such dependence would not appear in the stationary transition graph shown in **(a)**.

- the state space $S$ is spanned as $S = S^1 \times \cdots \times S^{d_S}$ (each state variable $S^k$ is called a factor), such that every state $s \in S$ is a $d_S$-dimension vector $s = (s^1, \ldots, s^{d_S})$.
- the space of observed states $O \subseteq S$ is denoted as $O = O^1 \times \cdots \times O^{d_O}$ with $d_O \le d_S$.
- the space of hidden states $H \subseteq S$ is spanned as $H = H^1 \times \cdots \times H^{d_H}$ with $d_H \le d_S$.
- $O \cup H = S$, $O \cap H = \emptyset$, such that $s = (o, h)$.
- $A$ is the set of actions $a$.
- $T(s_{t+1} \mid s_t, a_t) = \prod_{j=1}^{d_H} p(h_{t+1}^j | s_t, a_t) \prod_{i=1}^{d_O} p(o_{t+1}^i | s_t, a_t)$ is the transition probability function.
- $R(s_t, a_t, s_{t+1})$ is the reward function
- $\bar{O}(o_t \mid s_t)$ is the observation probability function that outputs 1 if $o_t \in O$ is subvector of $s_t \in S$ and 0 otherwise.

**Representing stochasticity in transitions as independent exogenous noise.** Via the Reparameterization Lemma (Appendix B of Buesing et al. (2018)), we can always reparameterize the stochasticity to be exogenous, and write the probabilistic transition of factored state variables in a factored-POMDP as a Structural Causal Model (SCM)

$$s_{t+1}^i := f_i(\mathbf{PA}_{s_{t+1}^i}, a_t, \epsilon_t^i), \quad i = 1, \ldots, d_S \tag{1}$$

where each $f_i$ represents an arbitrary deterministic function. $\mathbf{PA}_{s_{t+1}^i}$ denotes the set of parent state factors at time $t$, of $s_{t+1}^i$, such that there exists an edge from each element $s_t^j \in \mathbf{PA}_{s_{t+1}^i}$ to $s_{t+1}^i$ in the transition graph $\mathcal{G}$. Action $a_t$ is represented separately for clarity. The exogenous noise variable $\epsilon_t^i$ for each factor $i$ is jointly independent at each time step $t$, that is $p_{\epsilon_t^1, \ldots, \epsilon_t^{d_S}} = \prod_{i=1}^{d_S} p_{\epsilon_t^i}$. This noise variable can be seen as introducing stochasticity in the transitions, such that every $s_{t+1}^i = f_i(\mathbf{PA}_{s_{t+1}^i}, a_t^i, \epsilon_t^i)$ is a sample drawn from $p(s_{t+1}^i | \mathbf{PA}_{s_{t+1}^i}, a_t^i)$, for every $\epsilon_t^i$, consistent with the reparameterization lemma (Buesing et al., 2018). Thus, in Fig. 1, we can represent all stochasticity in transitions with independent exogenous noise nodes.

From the perspective of an agent, the uncertainty in predicting the next observables $o_{t+1}^i$ from the current observables and action, in a factored-POMDP setting, arises from two sources: the effect of current values of hidden factors $h_t$ and the unobservable stochasticity in the transition encapsulated by the current noise $\epsilon_t^i$. Therefore, if we somehow had access to the current hidden states $h_t$ and the noise $\epsilon_t^i$, then each next

state $s_{t+1}^i$ would be deterministically predictable given the current observed states $o_t$ and action $a_t^i$. For our factored-POMDP, similar to examples in real life, both $h_t$ and $\epsilon_t^i$ are not observable.

## 3 Deriving the algorithm for learning the transition dynamics of factored-POMDPs

In this section, we derive the DVAE-based framework for learning the transition model via approximate maximum likelihood. Since the transition at each step depends on current hidden factors, we first need to identify these hidden factors $\{h_t^i\}_{i=1}^{d_H}$ and then learn the factor-wise 1-step transition probabilities $p(s_{t+1}^j|\{o_t^i\}_{i=1}^{d_O}, \{h_t^i\}_{i=1}^{d_H}, a_t)$ for $j = 1, \ldots, d_S$.

In subsection 3.1, we demonstrate that the joint probabilities $p(o_{1:T}, h_{1:T}|a_{1:T})$ and $p(h_{1:T}|o_{1:T}, a_{1:T})$ can be factorized into products of conditional probabilities along time steps and state factors, which correspond to the factor-wise transition probabilities and the hindsight-based encoders for the hidden factors, respectively. In subsection 3.2, we derive the variational lower bound (VLB) within the DVAE framework, enabling the end-to-end learning of the parameterized hidden encoders (inference distribution) $q_\phi(h_t^i|h_{t-1}, o_{t-1:T}, a_{t-1:T})$ for $i = 1, \ldots, d_H$, as well as the transition models (generative distribution) $p_{\theta_o}(o_{t+1}^j|s_t, a_t)$ for $j = 1, \ldots, d_O$ and $p_{\theta_h}(h_{t+1}^j|s_t, a_t)$ for $j = 1, \ldots, d_H$. Subsection 3.3 introduces the conditional mutual information (CMI) metric for estimating the edges of the transition causal graph. In subsection 3.4, we extend the VLB derived in subsection 3.2 by incorporating additional masked loss terms, used to evaluate the CMI, and causal loss terms to refine the causal parents of each factor. Finally, subsection 3.5 presents our Modulo environment—a factored-POMDP example—to illustrate our results.

### 3.1 Factorization of joint distributions along time steps and state factors

We begin by factorizing the joint distribution $p(o_{1:T}, h_{1:T}|a_{1:T})$ over the time sequence into distinct time and state factors. This is achieved by leveraging the Markovian property and other d-separation conditions, under the assumptions of a factorized state space (outlined in the first four points of the Factored-POMDP definition) and factorized transition dynamics (as specified in the sixth point of the Factored-POMDP definition).

$$p(o_{1:T}, h_{1:T}|a_{1:T}) = \prod_{t=0}^{T-1} p(o_{t+1}, h_{t+1}|o_{1:t}, h_{1:t}, a_{1:T}) = \prod_{t=0}^{T-1} p(o_{t+1}|o_{1:t}, h_{1:t+1}, a_{1:T}) \, p(h_{t+1}|o_{1:t}, h_{1:t}, a_{1:T})$$

$$= \prod_{t=0}^{T-1} p(o_{t+1}|o_t, h_t, a_t) \, p(h_{t+1}|o_t, h_t, a_t) = \prod_{t=0}^{T-1} \left(\prod_{i=1}^{d_O} p(o_{t+1}^i|s_t, a_t)\right) \left(\prod_{j=1}^{d_H} p(h_{t+1}^j|s_t, a_t)\right) \tag{2}$$

Here, the first equation is obtained by applying the chain rule to decompose the joint distribution into a product of conditionals at each time step. Next, Bayes' rule is used to separately obtain the conditionals for the observation and hidden state (the second equation). In the third equation, we leverage the fact that $p(o_{t+1}|o_{1:t}, h_{1:t+1}, a_{1:T}) = p(o_{t+1}|o_t, h_t, a_t)$, as illustrated in Fig. 1c. In this expression, conditioning on $\{o_t, h_t, a_t\}$ removes the dependency of $o_{t+1}$ on the past information $\{o_{1:t-1}, h_{1:t-1}, a_{1:t-1}\}$ by blocking all paths from the past to $o_{t+1}$ (by virtue of the Markovian property). Furthermore, the dependency of $o_{t+1}$ on next hidden state $h_{t+1}$ is eliminated due to the blocked forks ($o_{t+1} \leftarrow o_t \rightarrow h_{t+1}$, $o_{t+1} \leftarrow h_t \rightarrow h_{t+1}$) when conditioned on their common parents $o_t$ and $h_t$. The dependency on future actions $a_{t+1:T}$ is similarly blocked by the unconditioned colliders on $o_{t+2:T}$ (e.g., $o_{t+1} \rightarrow o_{t+2} \leftarrow a_{t+1}$). Analogous d-separation arguments apply to the factorization of the hidden state transition in the third equation. Finally, by expressing $o_t = (o_t^1, \ldots, o_t^{d_O})$, $h_t = (h_t^1, \ldots, o_t^{d_H})$ and $s_t = (o_t, h_t)$ and applying factorized transition dynamics, we obtain the final equation. Here, the decomposed terms $p(h_{t+1}^j|s_t, a_t)$ and $p(o_{t+1}^i|s_t, a_t)$ represent the transition probabilities for the $j$-th hidden state and the $i$-th observed state, respectively.

Similarly, we factorize the posterior distribution $p(h_{1:T}|o_{1:T}, a_{1:T})$ as follows:

$$p(h_{1:T}|o_{1:T}, a_{1:T}) = \prod_{t=0}^{T-1} p(h_{t+1}|h_{1:t}, o_{1:T}, a_{1:T}) = \prod_{t=0}^{T-1} p(h_{t+1}|h_t, o_{t:T}, a_{t:T}) = \prod_{t=0}^{T-1} \prod_{j=1}^{d_H} p(h_{t+1}^j|h_t, o_{t:T}, a_{t:T})$$

(3)

Here, the first equation is obtained again by applying the chain rule. In the second equation, the influence of past information $\{h_{1:t-1}, o_{1:t-1}, a_{1:t-1}\}$ on $h_{t+1}$ is removed by conditioning on $\{h_t, o_t\}$ (Fig. 2c), thanks to the Markovian property. However, the dependency on future observations $o_{t+1:T}$ persists due to the unblocked chains from each future observation to $h_{t+1}$ (e.g., $h_t \rightarrow o_{t+1}$, $h_t \rightarrow h_{t+1} \rightarrow o_{t+2}$, etc.). Likewise, the dependency on future actions $a_{t+1:T}$ remains because of the paths created by conditioned colliders on the future observations $o_{t+2:T}$ (e.g., $h_{t+1} \rightarrow o_{t+2} \leftarrow a_{t+1}$, $h_{t+1} \rightarrow h_{t+2} \rightarrow o_{t+3} \leftarrow a_{t+2}$, etc.). The further factorization over state, as shown in the third equation, holds if any two hidden factors $h_{t+1}^i$ and $h_{t+1}^j$, (for $1 \le i < j \le d_H$) do not collide on any future observation factor $o_\tau^j$, (for $1 \le j \le d_O$ and $\tau \ge t+2$). In other words, all paths connecting $h_{t+1}^i$ and $h_{t+1}^j$ must be blocked, so that $h_{t+1}^i \perp\!\!\!\perp h_{t+1}^j|\{h_t, o_{t:T}, a_{t:T}\}$.

Note $p(h_t^j|h_{t-1}, o_{t-1:T}, a_{t-1:T})$ serves as an encoder for the $j$-th hidden state, given the conditioned Markovian state-action $h_{t-1}, o_{t-1}, a_{t-1}$, current and all future observations and action $o_{t:T}, a_{t:T}$, which we refer to as the *DVAE-based hindsight encoder*.

## 3.2 Learning the hidden encoder and transition dynamics via variational lower bound of DVAE

We aim to maximize the conditional marginal log-likelihood of the observations $o_{1:T}$ given the actions $a_{1:T}$, parameterized by $\theta$, under the true data distribution $p(o_{1:T}|a_{1:T})$:

$$\max_\theta \mathbb{E}_{p(o_{1:T}|a_{1:T})} \left[ \log p_\theta(o_{1:T}|a_{1:T}) \right]$$

(4)

By introducing a variational distribution $q_\phi(h_{1:T}|o_{1:T}, a_{1:T})$, parameterized by $\phi$, we can decompose the objective in Eq. 4 as follows (see Appendix A.1 for derivation):

$$\max_{\theta, \phi} \mathbb{E}_{p(o_{1:T}|a_{1:T})}[\ell_{\text{VLB}}(\theta, \phi; o_{1:T}, a_{1:T}) + D_{\text{KL}}(q_\phi(h_{1:T}|o_{1:T}, a_{1:T}) \| p_\theta(h_{1:T}|o_{1:T}, a_{1:T}))]$$

(5)

Here, $\ell_{\text{VLB}}(\theta, \phi; o_{1:T}, a_{1:T})$ is the variational lower bound (VLB) on the marginal log-likelihood, serving as a lower bound due to the non-negativity of the KL divergence term. VLB is defined as:

$$\ell_{\text{VLB}}(\theta, \phi; o_{1:T}, a_{1:T}) = \mathbb{E}_{q_\phi(h_{1:T}|o_{1:T}, a_{1:T})} \left[ \log p_\theta(o_{1:T}, h_{1:T}|a_{1:T}) - \log q_\phi(h_{1:T}|o_{1:T}, a_{1:T}) \right]$$

(6)

Thus, optimizing Eq. 4 reduces to maximizing the expected VLB. In practice, we approximate the expectation of the data distribution $p(o_{1:T}|a_{1:T})$, using observed data sequences. We employ independent and identically distributed (i.i.d.) sampled trajectories from the collected dataset $\mathcal{D}$ to construct a Monte Carlo estimate of the expected VLB, defined as follows:

$$\mathcal{L}_{\text{VLB}}(\theta, \phi; o_{1:T}, a_{1:T}) = \mathbb{E}_{(o_{1:T}, a_{1:T}) \sim \mathcal{D}} \left[ \ell_{\text{VLB}}(\theta, \phi; o_{1:T}, a_{1:T}) \right]$$

(7)

By applying the factorization across time steps and state factors as described in Eqs. 2 and 3 to the generative model $p_\theta$ and the inference model $q_\phi$ in VLB of Eq. 6 respectively, we obtain the following VLB (see Appendix A.2 for details):

$$\ell_{\text{VLB}}(\theta, \phi, \overline{\phi}; o_{1:T}, a_{1:T}) = \sum_{t=0}^{T-1} \mathbb{E}_{q_\phi(h_{1:t}|o_{1:T}, a_{1:T})} \left[ \sum_{j=1}^{d_O} \log p_{\theta_o}(o_{t+1}^j|s_t, a_t) \right.$$
$$\left. - \sum_{j=1}^{d_H} D_{\text{KL}}(q_{\overline{\phi}}(h_{t+1}^j|h_t, o_{t:T}, a_{t:T}) \| p_{\theta_h}(h_{t+1}^j|s_t, a_t)) \right]$$

(8)

Here, $\theta = \theta_o \cup \theta_h$ represents the parameters of the generative model. $q_{\overline{\phi}}^-(h_{t+1}^j|h_t, o_{t:T}, a_{t:T})$ serves as the target distribution of the next encoded hidden state in the KL divergence term, comparing it to the distribution of the next predicted hidden state $p_{\theta_h}(h_{t+1}^j|s_t, a_t)$. The notation $\overline{\phi}$ denotes the stop-gradient version of $\phi$, which is detached from the computation graph and replaced by a copy of $\phi$ from the previous training step. Using a stop-gradient target in self-predictive representations is common in practice (Zhang et al., 2020; Ghugare et al., 2022), as this technique helps avoid representational collapse (Ni et al., 2024).

*Remark* 1 (**DVAE-based hindsight encoder for inferring the current hiddens**). *Eq. ?? shows that the joint conditional of the hidden states can be decomposed into $T$ conditionals, each conditioned on 1-step past, current and all future observations and actions, as well as the 1-step past hidden states. The previous hidden states are recursively chained across the $T$ time steps, effectively incorporating the entire past. Thus, the hidden encoder $q_\phi(h_{t+1}^j|h_t, o_{t:T}, a_{t:T})$ systematically leverages all available information to infer the distribution of hidden states.*

*Remark* 2 (**Identifiability of current hiddens**). *The hidden states need to be identified by our encoder to learn the causal dynamics. Informally paraphrasing Khemakhem et al. (2020) and Hyvärinen et al. (2024), in the VAE setting, if (i) the prior over the hidden states is conditionally independent given certain observed variables, and (ii) the generated observations can be expressed as the sum of a bijective function of the hidden state and an independent noise variable, then the encoded hidden variable is identifiable up to a simple transformation. In our experimental results below, we demonstrate this identifiability result for standard VAEs also in our DVAE framework for a 1-hidden variable factored-POMDP case, by empirically showing that our encoder's identified hidden state corresponds to an invertible linear transformation of the true hidden state (see also Table 1).*

*Remark* 3 (**History-based encoder vs. DVAE-based hindsight encoder**). *A history-based encoder $q_\phi(h_t|o_{1:t}, a_{1:t})$, which conditions only on past and current observations and actions, cannot fully infer the current hidden state because it depends on an exogenous noise variable independent of these inputs (Fig. 1). This limitation, known as the conditioning gap, arises from incomplete information—specifically, ignoring future observations needed to disambiguate distinct hidden states (Bayer et al., 2021; Becker & Neumann, 2022). In contrast, our DVAE-based hindsight encoder leverages future observations (which carry essential information about this noise) to refine its inference.*

*Remark* 4 (**Current and 1-step hindsight encoder vs. DVAE-based hindsight encoder**). *Rewriting Eq. 1 as $o_{t+1}^i = f_i(o_t, h_t^j, a_t, \epsilon_t^i)$, for every $i$ and $j$, shows that an encoder conditioned on $o_t, a_t$, and $o_{t+1}^i$ (the current and 1-step hindsight encoder as in Jarrett et al. (2023)) would infer $h_t^j$ by inverting the transition function of the parent of $h_t^j$, i.e., $o_{t+1}^i$. However, the inferred $h_t^j$ would be contaminated with $\epsilon_t^i$. Indeed, Jarrett et al. (2023) exclude any hidden states in their environment, encoding only the exogenous noise from current and 1-step future data. Our DVAE-based hindsight encoder, on the other hand, uses additional past information, the bootstrapped 1-step past hidden state, and multiple future observations and actions to better disentangle the current hidden state from the exogenous noise.*

*Remark* 5 (**Full trajectory bidirectional-RNN-based encoder vs. DVAE-based hindsight encoder**). *A bidirectional RNN encoder $q_\phi(h_t|o_{1:T}, a_{1:T})$, conditioned on the complete trajectory, simply compresses past and future observations and actions into the current hidden representation, without taking into account inferred hiddens at previous time points. In contrast, the DVAE-based hindsight encoder takes a more principled approach, as derived from the DVAE framework, by using an additional bootstrapped hidden state, recursively constructed from the full trajectory including previously inferred hiddens, to encode the current hidden representation.*

## 3.3 Transition Graph Estimation

The causal dependency of each transition pair $s_t^i \to s_{t+1}^j$ or $a_t \to s_{t+1}^j$ is estimated through conditional mutual information (CMI) (Wang et al., 2022). During evaluation, the CMI is computed based on two learned transition distributions: the full transition model $p_\theta(s_{t+1}^j|s_t, a_t)$, which leverages all state variables and the action to predict the next state of the $j$-th factor, and the masked transition model $p_\theta(s_{t+1}^j|s_t \setminus s_t^i, a_t)$, which relies on all state factors except for $s_t^i$ for prediction.

Specifically, when the next state $s_{t+1}^j$ is observable (denoted as $o_{t+1}^j$), the $\text{CMI}^{i,j}$ between $s_t^i$ and $o_{t+1}^j$ given $\{s_t \backslash s_t^i, a_t\}$ is formulated as:

$$I(s_t^i; o_{t+1}^j | s_t \backslash s_t^i, a_t) = \mathbb{E}_{s_t, a_t, o_{t+1}^j \sim \mathcal{D}, q_\phi} \left[ \log \frac{p_{\theta_o}(o_{t+1}^j | s_t, a_t)}{p_{\theta_o}(o_{t+1}^j | s_t \backslash s_t^i, a_t)} \right] \tag{9}$$

Here, $s_t^i$ can be either an observed state or a hidden state sampled from the hidden encoder. The expectation in the CMI is approximated by aggregating transitions from all episodes in a mini-batch.

When the next state $s_{t+1}^j$ is hidden (denoted as $h_{t+1}^j$), the $\text{CMI}^{i,j}$ between $s_t^i$ and $h_{t+1}^j$ conditioned on $\{s_t \backslash s_t^i, a_t\}$ is given by:

$$I(s_t^i; h_{t+1}^j | s_t \backslash s_t^i, a_t) = \mathbb{E}_{s_t, a_t \sim \mathcal{D}, q_\phi} \left[ D_{\text{KL}}(p_{\theta_h}(h_{t+1}^j | s_t, a_t) \, \| \, p_{\theta_h}(h_{t+1}^j | s_t \backslash s_t^i, a_t)) \right] \tag{10}$$

The derivations of Eqs. 9 and 10 are provided in Appendix A.3. Note that for causal dependency between the action and the next state, $a_t \to s_{t+1}^j$, the same CMI formula applies by replacing $s_t^i$ with $a_t$ in the conditioning set, which then becomes $\{s_t\}$.

In practice, the existence of an edge in the transition graph, i.e., $s_t^i \to s_{t+1}^j$ or $a_t \to s_{t+1}^j$, is determined by whether the corresponding CMI value $\text{CMI}^{i,j}$ exceeds a predefined threshold $\delta$. The binarized CMI matrix is then applied to select the parents of each next state in the causal transition losses in Eq. 11, and thus, refines learning of the causal transition dynamics $p_\theta(s_{t+1}^j | \mathbf{PA}_{s_{t+1}^j})$.

### 3.4  Adding extra loss terms to learn masked transitions (for CMI) and to refine causal edges

To evaluate the conditional mutual information (CMI) in Eqs. 9 and 10, we introduce additional loss terms into Eq. 8, which currently comprises only the factor-wise (full) transition terms $p_{\theta_o}(o_{t+1}^j | s_t, a_t)$ and $p_{\theta_h}(h_{t+1}^j | s_t, a_t)$. Inspired by (Wang et al., 2022), we incorporate extra masked and causal losses into Eq. 8. Specifically, in addition to the full transition distribution (without masking any input factor), we compute a masked transition distribution by masking a randomly chosen state factor $s_t^i$ or action $a_t$ from each input $\{s_t, a_t\}$ to the transition model, and also a causal transition distribution by masking out all input factors except for causal parents of $s_{t+1}^j$ identified using the transition graph learned so far. These are used in the 3 loss types, for both the Negative Log-Likelihood (NLL) of observed states and the KL-divergence (KL-div) of hidden states, to yield 6 loss terms:

$$\ell_{\text{VLB}}\left(\theta, \phi, \overline{\phi}; o_{1:T}, a_{1:T}\right) = \sum_{t=0}^{T-1} \mathbb{E}_{q_\phi(h_{1:t} | o_{1:T}, a_{1:T})} \left[ -\sum_{j=1}^{d_O} \Big[ \underbrace{-\log p_{\theta_o}(o_{t+1}^j | s_t, a_t)}_{\text{Full NLL Loss}} \underbrace{-\log p_{\theta_o}(o_{t+1}^j | s_t \backslash s_t^i, a_t)}_{\text{Masked NLL Loss}} \right.$$

$$\underbrace{-\log p_{\theta_o}(o_{t+1}^j | \mathbf{PA}_{o_{t+1}^j})}_{\text{Causal NLL Loss}} \Big] - \sum_{j=1}^{d_H} \Big[ \underbrace{D_{\text{KL}}(q_{\overline{\phi}}(h_{t+1}^j | h_t, o_{t:T}, a_{t:T}) \, \| \, p_{\theta_h}(h_{t+1}^j | s_t, a_t))}_{\text{Full KL-Div Loss}}$$

$$+ \underbrace{D_{\text{KL}}(q_{\overline{\phi}}(h_{t+1}^j | h_t, o_{t:T}, a_{t:T}) \, \| \, p_{\theta_h}(h_{t+1}^j | s_t \backslash s_t^i, a_t))}_{\text{Masked KL-Div Loss}} + \underbrace{D_{\text{KL}}(q_{\overline{\phi}}(h_{t+1}^j | h_t, o_{t:T}, a_{t:T}) \, \| \, p_{\theta_h}(h_{t+1}^j | \mathbf{PA}_{h_{t+1}^j}))}_{\text{Causal KL-Div Loss}} \Big] \Big] \tag{11}$$

Here, the network parameters $\theta_o, \theta_h$ and $\phi$ $(\overline{\phi})$ are shared across all three types of losses. $s_t \backslash s_t^i = \{s_t^1, \ldots, s_t^{i-1}, s_t^{i+1}, \ldots, s_t^{d_S}\}$ denotes the set of all state factors at time $t$ except for the $i$-th factor $s_t^i$. For each factor $j$ in each sample within every mini-batch, the index $i$ is sampled independently and identically from a uniform distribution over $\{1, \ldots, d_S\}$. The term $\mathbf{PA}_{s_{t+1}^j}$ are inferred from the learned transition graph so far using the conditional mutual information between each pair of factors, as discussed in Section 3.3.

Specifically, the masked NLL and KL-div losses serve as training regularizers and approximate the factor-wise masked transition terms $p_{\theta_o}(o_{t+1}^j | s_t \backslash s_t^i, a_t)$ and $p_{\theta_h}(h_{t+1}^j | s_t \backslash s_t^i, a_t)$, which are essential for computing

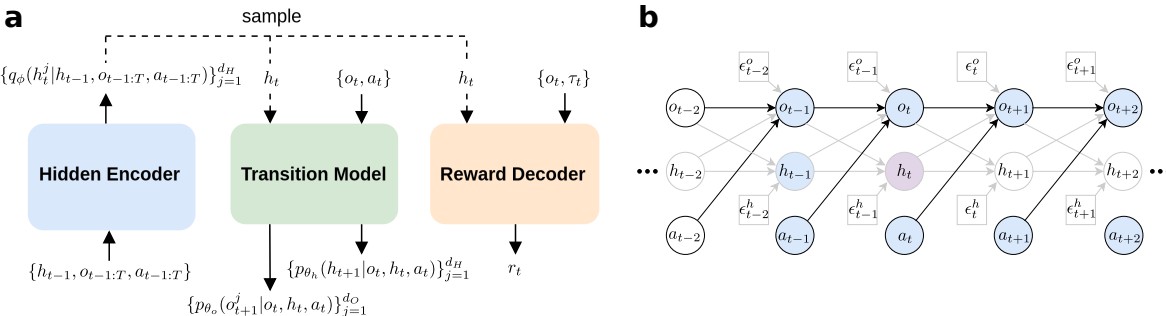

Figure 2: **(a)** Model architecture for computing the objective function in Eq. 13. **(b)** The current hidden state (light purple) is inferred from the hidden states, observables, and action of the previous step, along with the current and all future observables and actions (light blue) within the DVAE-based hindsight encoder.

the CMI as described in section 3.3. The causal NLL and KL-divergence losses are used to learn the causal transition dynamics $p_{\theta_o}(o_{t+1}^j|\mathbf{PA}_{o_{t+1}^j})$ and $p_{\theta_h}(h_{t+1}^j|\mathbf{PA}_{h_{t+1}^j})$, where conditioning is performed solely on identified causal parents rather than on the full set $\{s_t, a_t\}$.

Finally, the KL-div term in Eq. 11, which aligns the distributions of the encoded and predicted next hidden representations, can drive convergence toward a trivial constant hidden representation $c$, i.e., $p_{h_\theta}(h_{t+1}^j|s_t, a_t) = q_{\overline{\phi}}(h_{t+1}^j|h_t, o_{t:T}, a_{t:T}) = c$ (Ni et al., 2024). This problem arises from the bootstrapped nature of hidden representations in self-predictive learning—a phenomenon akin to posterior collapse in VAEs (He et al., 2019), though occurring in a different context. To prevent such degeneration, additional constraints on the hidden representation need to be applied alongside the VLB. Here, we employ a reward predictor parameterized by $\psi$ to condition the encoded hidden representations $h_{1:T}$, trained by minimizing the prediction error:

$$\mathcal{L}_{\text{rew}}\left(\phi, \psi; o_{1:T}, \tau_{1:T}, r_{1:T}\right) = \mathbb{E}_{\substack{(o_{1:T}, \tau_{1:T}, r_{1:T}) \sim \mathcal{D} \\ h_{1:T} \sim q_\phi(h_{1:T}|o_{1:T}, a_{1:T})}}\left[\ell_{\text{rew}}\left(\psi; o_{1:T}, h_{1:T}, \tau_{1:T}, r_{1:T}\right)\right] \tag{12}$$

Here, $\tau_t$ denotes any reward-related variables (e.g., a time-dependent/episodic target) used to predict the reward accurately. In our experiment, $\tau_t$ is defined as an episodic target state, whose components are observable and hidden factors. These components are randomly sampled at the beginning of each episode and remain fixed throughout the episode. The reward $r_t$ is defined as the number of matching factors between the current state $s_t = (o_t, h_t) = (o_t^1, \ldots, o_t^{d_O}, h_t^1, \ldots, o_t^{d_H})$ and the target state $\tau_t = (\tau_t^1, \ldots, \tau_t^{d_O+d_H})$. Consequently, the supervised reward loss imposes additional constraints on the encoded hidden state $h_t$, preventing representational collapse. An illustrative example of the episodic target state $\tau_t$ is the desired health outcome of a patient, indicating recovery, which medical interventions aim to achieve. The reward can then be defined according to how closely the patient's current health state matches this target state. $\ell_{\text{rew}}$ can be any supervised loss function; in our experiments, we use cross-entropy loss for categorical rewards that takes discrete values.

Combining all components, we obtain the final objective to be minimized. This objective is a weighted sum of the mean VLB from Eqs. 7 and 11, and mean reward loss from Eq. 12, with a weight coefficient $\lambda > 0$:

$$\mathcal{L}_{\text{obj}}\left(\theta, \phi, \overline{\phi}, \psi; o_{1:T}, a_{1:T}, \tau_{1:T}, r_{1:T}\right) = -\mathcal{L}_{\text{VLB}}\left(\theta, \phi, \overline{\phi}; o_{1:T}, a_{1:T}\right) + \lambda \mathcal{L}_{\text{rew}}\left(\phi, \psi; o_{1:T}, \tau_{1:T}, r_{1:T}\right) \tag{13}$$

The model architecture depicted in Fig. 2a illustrates that every hidden states $h_t^j$ is obtained through temporally recursive sampling from $q_\phi(h_\tau^j|h_{\tau-1}, o_{\tau-1:T}, a_{\tau-1:T})$ for $\tau = 1$ to $t$. Then, the hidden sample at each time step $t$ is fed into the transition model and reward decoder to predict next states and reward. The unrolled probabilistic transition graph in Fig. 2b highlights the temporal data used as inputs to the DVAE-based hindsight encoder for the hidden states. The full details of the algorithm are provided in Appendix A.4.

### 3.5 Modulo environment: a stochastic, discrete state-action, factored-POMDP

Modified from Ke et al. (2021), we construct a probabilistic discrete Factored-POMDP environment, to examine the performance of our model on inferring the hidden states and underlying transition graph. We called this environment modulo environment as the modulo operator is involved in its transition dynamics defined as $s_{t+1} := (As_t + a_t + \epsilon_{t+1}) \mod l$, where $l$ denotes the number of possible discrete values and $A$ is the adjacency matrix of the transition graph $\mathcal{G}$. At time step $t$, each discrete factor $s_t^i$ of the state vector $s_t = (s_t^1, \ldots, s_t^{d_S})^\top$ has values within $\{0, \ldots, l-1\}$, the binary element $a_t^i$ of the action vector $a_t = (a_t^1, \ldots, a_t^{d_S})^\top$ represents if the $i$-th factor is intervened or not by setting $a_t^i = 1$ or 0 respectively, and the noise vector $\epsilon_t = (\epsilon_t^1, \ldots, \epsilon_t^{d_S})^\top \in E$ is sampled from a jointly independent distribution $p_{\epsilon_t} = \prod_{i=1}^{d_S} p_{\epsilon_t^i}$. Fig. 3 depicts noise-free transition dynamics with different underlying transition graph structures.

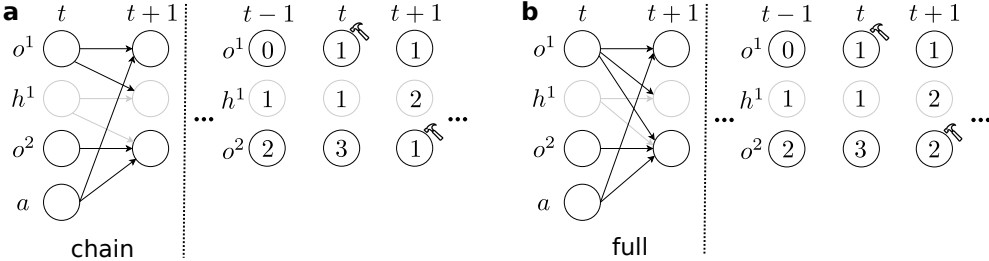

Figure 3: Illustration of Modulo environment with different types of transition graphs that have $d_S = 3$ and $l = 4$. **(a)** Chain structure. left: ground truth transition graph, right: next states depend on current states and action. **(b)** Same demonstration for the full structured (lower-triangular adjacency matrix) transition graph. The hammer symbol denotes the action intervened on any of the observed states at each time step.

Our environment satisfies two properties. **(P1)** For every hidden factor $h_t^i$, there exists at least one observable $o_{t+1}^j$, such that $h_t^i \in \mathbf{PA}_{o_{t+1}^j}$. **(P2)** The transition map $f \equiv \{f_i\}_{i=1}^{d_S}$ in Eq. 1, for every $a \in A$ and every $\epsilon \in E$, i.e. $(f)_{a,\epsilon} : S \to S$ from any $s_t \in \mathcal{S}$ to $s_{t+1} \in \mathcal{S}$, is bijective, where $s$ is the full state with all observable and hidden factors. For property P1, various transition graph structures are permissible, provided that at least one child factor at time step $t+1$ of the hidden factor at time step $t$ is observable. For property P2, given that the state and action spaces are discrete and the transition dynamics involve an element-wise modulo operation applied to a linear combination of the current state-action pair and noise—specifically, $s_{t+1} := (As_t + a_t + \epsilon_{t+1}) \mod l$—the mapping from $s_t$ to $s_{t+1}$ is always bijective for any fixed $a_t$ and $\epsilon_t$.

Indeed, by assuming a version of **(P2)**, in an environment with only exogenous noise but no hidden factors, we can deterministically infer these exogenous noise variables at $t$, by using a current and 1-step hindsight encoder for the hiddens similar to the latent generator in Jarrett et al. (2023), which learns to invert $f$ using observables and action at current $t$ and observables at 1-step future $t+1$. However, *with both hidden factors and exogenous noise, despite these simplifying properties, history-based, and current and 1-step hindsight-based approaches are unable to learn the hidden factor and the graph*, as shown by the following experiments (see also Remark 3).

We also assume that the initial hidden state remains fixed across episodes, while the initial observed states are randomly sampled in each episode. A practical example illustrating our assumption of fixed initial hidden states is the medical scenario modeled by a factored-POMDP, where a patient's initial latent health factors are confined to a narrow range and subsequently evolve over time, influenced by medical interventions and disease progression.

## 4 Experiments demonstrate the effectiveness of DVAE-based hindsight encoder

**Environment setting.** We consider a straightforward yet non-trivial setup using the modulo environment: a chain-structured transition graph with $d_S = 3$ and $l = 4$, with 3 factors: an observable $o^1$, a middle hidden state $h^1$, then an observable $o^2$. The environment includes a stationary discrete noise distribution defined as $p(\epsilon_t^i = -1) = p(\epsilon_t^i = 1) = 0.05$ and $p(\epsilon_t^i = 0) = 0.9$ for $i = 1, 2, 3$. The principles outlined here can be extended

to other graph structures and larger values of $d_S$ and $l$, as empirically demonstrated later. Specifically, the transition dynamics in this setup are defined as $o_{t+1}^1 := (o_t^1 + a_t^1 + \epsilon_t^1) \bmod 4$, $h_{t+1}^1 := (o_t^1 + h_t^1 + \epsilon_t^2) \bmod 4$, and $o_{t+1}^2 := (h_t^1 + o_t^2 + a_t^3 + \epsilon_t^3) \bmod 4$. We assume that both the number and the dimension $l$ of the discrete hidden factors are known in advance. A random policy is employed to collect offline trajectories. In our modulo environment setting, randomly chosen actions sufficiently explore the state-action space, ensuring that the offline dataset adequately covers most state-action transitions.

**Baselines and our DVAE encoders.** We compare the performance of 6 different hidden encoders, each learned end-to-end with the same transition model and reward predictor architecture:

- **History Enc.** A history-based encoder, using complete past and current observations and actions: $q_\phi(h_t|o_{1:t}, a_{1:t})$, parameterized by a forward RNN.

- **Current & 1-Step Hindsight Enc.** A current and 1-step hindsight encoder (Jarrett et al., 2023), using current observations and action, and next step future observations: $q_\phi(h_t|o_{t:t+1}, a_{t:t+1})$, parameterized by an MLP.

- **Current & Full Hindsight Enc.** A current and full hindsight encoder, using current and all future observations and actions: $q_\phi(h_t|o_{t:T}, a_{t:T})$, parameterized by a backward RNN.

- **Full Trajectory Enc.** A full trajectory encoder, using the entire trajectory of observations and actions: $q_\phi(h_t|o_{1:T}, a_{1:T})$, parameterized by a forward RNN combining the past and current data and a backward RNN combining the future data, whose outputs are themselves combined by an MLP to yield $h_t$.

- **DVAE 1-Step Hindsight Enc.** A DVAE-based encoder with 1-step hindsight, using 1-step past (including sampled hidden), current, and 1-step future observations and actions: $q_\phi(h_t|h_{t-1}, o_{t-1:t+1}, a_{t-1:t+1})$.

- **DVAE Full Hindsight Enc.** A DVAE-based encoder with full hindsight, using 1-step past (including sampled hidden), current, and all future observations and actions: $q_\phi(h_t|h_{t-1}, o_{t-1:T}, a_{t-1:T})$.

**Implementation details.** We use the Adam optimizer with a learning rate $\alpha = 5\mathrm{e}{-4}$. Details on the neural network parameterization of the hidden encoder, transition model, and reward predictor are provided in Appendix A.5. The hyperparameters for the transition model largely follow Wang et al. (2022), and the hidden encoder and reward decoder are initialized to be compatible with the transition model. The transition graph is updated and evaluated with CMI threshold $\delta = 0.03$ every $N = 200$ training steps, remaining fixed during each interval.

**Results: DVAE Hindsight Encoders outperform History, Current & Hindsight, and Full Trajectory Encoders.** In Fig. 4, we empirically compare the training performances and evaluated CMI matrices across 6 types of encoders under 2 settings with exogenous noise $\epsilon_t$ applied to either the hidden state transition (noisy hidden setting) or the observed state transition (noisy observation setting).

In the noisy hidden setting (Fig. 4a and b), encoders with hindsight information converge to zero loss for both observed state predictions (the full NLL term of the VLB in Eq. 11) and reward prediction (the cross-entropy loss in Eq. 12). These encoders also infer correct transition graphs after binarizing their evaluated CMI matrices using the threshold $\delta$. In contrast, the history-based encoder struggles to train effectively, resulting in a CMI matrix with values close to $\delta$, which reflects less statistical confidence in the existence of corresponding causal edges. Without access to the next observation $o_{t+1}$, the history-based encoder cannot deterministically infer the current hidden state $h_t$, given the unknown noise $\epsilon_{t-1}$ affecting the transition to $h_t$. However, hindsight-based approaches can learn $h_t$ by utilizing information from observed states, which serve as children of the hidden state in the transition graph, thereby enabling accurate learning of the transition graph. Due to the unobserved exogenous noise injected into the hidden state transition, the transition model can only predict the next hidden state in distribution. As a result, prediction losses for the hidden state (measured by the full KL divergence between the encoded and predicted next hidden states in Eq. 11) do not decrease to zero for all encoders.

In the noisy observation setting (Fig. 4c and d), the DVAE-based encoder successfully learns hidden representations, allowing it to accurately predict the next hidden states and rewards, while all the other types of

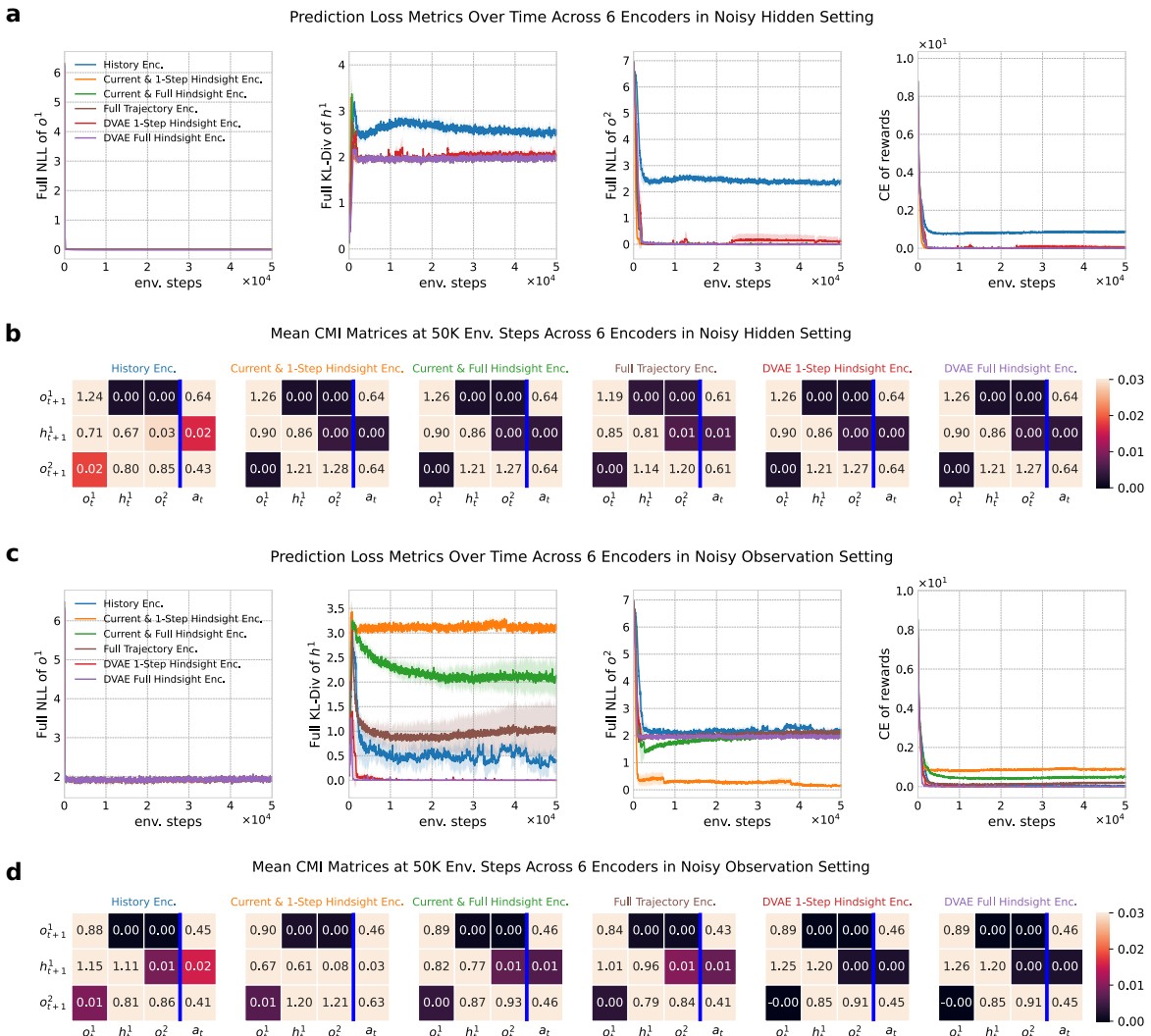

Figure 4: **(a)** Comparison of 6 encoder types, showing training profiles of state and reward prediction losses with mean and standard deviation (std) when noise $\epsilon_t^h$ is applied to the hidden state transition. **(b)** Corresponding mean CMI matrices evaluated at the end of training across the 5 encoders, displayed as heatmaps under the same conditions. Similarly, **(c)** and **(d)** present training performance and evaluated CMI matrices, respectively, when noise $\epsilon_t^h$ is applied to the observed state transitions. In all experiment results, each loss metric and CMI calculation for each encoder is run over 3 seeds. The color bar range is capped at the CMI threshold $\delta$, so that light color denotes an edge, and dark color no edge. The DVAE encoders produce CMI matrices whose binarized values match ground-truth. In Appendix B.1, we show the zoomed loss profiles of rows (a) and (c) to distinguish the encoders that overlap after convergence.

encoders fail to achieve similar performance (as seen in the second and fourth panels of Fig. 4c). In the third panel of Fig. 4c), it appears that the current and 1-step hindsight encoder achieves the lowest loss; however, this occurs because the encoder infers the hidden state $h_t^1$ as a direct copy of the observable $o_{t+1}^2$. Consequently, the transition model $o_{t+1}^2 = f_2(h_t^1, o_t^2, a_t^3, \epsilon_t^3)$ degenerates into a trivial identity mapping $o_{t+1}^2 = h_t^1$. This pathological solution effectively disregards the other conditioning factors $\{o_t^2, a_t^3, \epsilon_t^3\}$ and, consequently, results in incorrect edges within the estimated transition graph. Encoders other than DVAE-based encoders produce CMI matrices with values closer to threshold, or even infer spurious edges. We hypothesize that the DVAE-based model's ability to identify the current hidden state $h_t$ stems from its recursive structure (see Eq. 23), which combines sample-based past (Markovian) information with future information. In con-

| # Past | # Future | Graph | $h^1$ Decoding | $o^1, o^2$ Prediction | $h^1$ Prediction | Reward Prediction |
|---|---|---|---|---|---|---|
| | | | | Evaluation Accuracy in Noisy Hidden / Observation Setting | | |
| | | | | **History-Based Encoder** | | |
| all | 0 | $0.944_{(0.039)}$ / $1.000_{(0.000)}$ | $0.865_{(0.019)}$ / $0.971_{(0.021)}$ | $1.000_{(0.000)}$, $0.872_{(0.023)}$ / $0.915_{(0.017)}$, $0.876_{(0.012)}$ | $0.850_{(0.029)}$ / $0.964_{(0.026)}$ | $0.927_{(0.020)}$ / $0.997_{(0.002)}$ |
| | | | | **Current and Hindsight-Based Encoder** | | |
| 0 | 1 | $1.000_{(0.000)}$ / $0.944_{(0.079)}$ | $1.000_{(0.000)}$ / $0.866_{(0.044)}$ | $1.000_{(0.000)}$, $1.000_{(0.000)}$ / $0.915_{(0.017)}$, $0.996_{(0.003)}$ | $0.899_{(0.009)}$ / $0.814_{(0.009)}$ | $1.000_{(0.000)}$ / $0.949_{(0.005)}$ |
| 0 | all | $1.000_{(0.000)}$ / $1.000_{(0.000)}$ | $1.000_{(0.000)}$ / $0.914_{(0.017)}$ | $1.000_{(0.000)}$, $1.000_{(0.000)}$ / $0.915_{(0.017)}$, $0.897_{(0.007)}$ | $0.899_{(0.009)}$ / $0.870_{(0.031)}$ | $1.000_{(0.000)}$ / $0.959_{(0.009)}$ |
| | | | | **Full Trajectory-Based Encoder** | | |
| all | all | $1.000_{(0.000)}$ / $1.000_{(0.000)}$ | $1.000_{(0.000)}$ / $0.974_{(0.015)}$ | $1.000_{(0.000)}$, $1.000_{(0.000)}$ / $0.915_{(0.017)}$, $0.897_{(0.018)}$ | $0.899_{(0.009)}$ / $0.957_{(0.027)}$ | $1.000_{(0.000)}$ / $0.985_{(0.010)}$ |
| | | | | **DVAE-Based Hindsight Encoder** | | |
| all | 1 | $1.000_{(0.000)}$ / $1.000_{(0.000)}$ | $1.000_{(0.000)}$ / $1.000_{(0.000)}$ | $1.000_{(0.000)}$, $1.000_{(0.000)}$ / $0.915_{(0.017)}$, $0.905_{(0.009)}$ | $0.895_{(0.009)}$ / $1.000_{(0.000)}$ | $1.000_{(0.000)}$ / $1.000_{(0.000)}$ |
| all | all | $1.000_{(0.000)}$ / $1.000_{(0.000)}$ | $1.000_{(0.000)}$ / $1.000_{(0.000)}$ | $1.000_{(0.000)}$, $1.000_{(0.000)}$ / $0.915_{(0.017)}$, $0.905_{(0.009)}$ | $0.899_{(0.009)}$ / $1.000_{(0.000)}$ | $1.000_{(0.000)}$ / $1.000_{(0.000)}$ |

Table 1: Evaluation accuracies across various metrics, including transition graph accuracy (measured by the match between inferred and ground truth edges), hidden state decoding accuracy (linear decoding accuracy of encoded hidden states to ground truth hidden states), observation prediction accuracy, hidden state prediction accuracy (measured by the match between predicted and encoded next hidden states), and reward prediction accuracy. These metrics are reported for 6 types of encoders utilizing different steps of past and future observables in both noisy hidden and noisy observation settings. Each accuracy value is presented as mean$_{\text{std}}$ over 3 runs. Lavender and beige highlights indicate suboptimal accuracy values for certain encoders in the noisy hidden and observation settings, respectively. Note that the DVAE-based encoder is labeled as using all past observables, as it estimates the 1-step past hidden state based on recursive hidden samples from the beginning of an episode, which requires all past observables.

trast, the history-based and current hindsight-based encoders, which rely on a single directional view of observables along the trajectory, lack sufficient information to identify the current hidden state in the noisy observation setting. We show that even the encoder using the entire trajectory still underperforms compared to our DVAE-based encoder. By exploiting forward estimation through bootstrapping, our approach imposes a more structured, constrained way of encoding the hidden state, providing clear advantages over the trajectory-based encoder parameterized by bidirectional RNNs, despite both having access to the same data. Finally, similar to the prediction of the next state hidden in the noisy hidden setting, the transition model can only predict noisy observations in distribution.

We also tabulate the accuracy of graph edges, decoding of encoded hidden, and state transitions, after convergence, of the six encoder architectures across both noise settings, in Table 1. In the noisy hidden setting, the lower accuracies of the history-based encoder, highlighted in lavender, indicate its inability to learn the hidden state and accurately perform the corresponding transition and reward predictions. Ideally, the encoded hidden state should be linearly decodable to its ground truth value and deterministically predictive of the reward, as reflected by perfect $h^1$ decoding and reward prediction accuracy in all other encoders. Additionally, the expected $h^1$ prediction accuracy should be approximately 0.9, accounting for the 10% noise in the hidden transition, assuming both the encoded and predicted hidden states are optimally learned. Indeed, the mean $h^1$ prediction accuracy for all encoders, except the history-based one, is very close to 0.9.

Similarly, in the noisy observation setting, the accuracies highlighted in beige indicate suboptimal encoding and prediction of the hidden states for the history-based and current hindsight-based encoders. Interestingly, for the current and hindsight-based encoder, the mean $o^2$ prediction accuracy exceeds the expected value of 0.9 and approaches 1.0 (see also third panel of Fig. 4c), suggesting that this encoder copies its input of the next noisy $o^2$ as the hidden state. This copying approach, however, trades off accuracy in $h^1$ and reward prediction compared to encoders that do not learn this inconsequential solution for the hidden state. The DVAE-based encoders perform optimally in both noise settings. Here the perfect hidden state decoding accuracy indicates that the hidden state inferred by our DVAE-based encoders is identifiable up to an invertible linear transformation. The causal relationships associated with the linearly transformed hidden state can be effectively captured using the CMI metric, just as they are for the ground-truth hidden state. Notably, the DVAE 1-step Hindsight Encoder achieves the same optimal performance as the theoretically-derived DVAE Full Hindsight Encoder due to the absence of cascaded hidden factors in our environment.

We evaluate our DVAE 1-step Hindsight Encoder on different transition graph structures, which consistently recovers the true causal graphs (see Appendix B.2). We also test all baseline models in scenarios with a

larger state space and with two uncascaded hidden factors (see Appendices B.3 and B.4, respectively). In both cases, although none of baselines is able to perfectly identify the hidden states, likely due to the inherent challenges of gradient-based optimization in discrete state-action spaces (Niculae et al., 2023), our DVAE 1-step Hindsight Encoder still consistently achieves better performances than the others. The experimental results presented here serve as a proof of principle. Specifically, we focus on demonstrating the necessity of incorporating both past and future contexts in a principled manner for hidden state identification and causal transition learning. This work does not address the challenge of representation learning from high-dimensional observations, which will be discussed further in the following section.

## 5   Discussion

We have demonstrated that the proposed DVAE-based hindsight encoder effectively identifies hidden state factors and learns the causal transition graph in a factored-POMDP with 1 hidden state, outperforming both history-based and typical hindsight-based encoders. This approach shows particular promise in settings with access to full offline trajectories. Related works (Uehara et al., 2023; Zhang & Jiang, 2024) also incorporate future-dependent objectives in offline POMDPs, utilizing robust or pessimistic frameworks to handle multi-step constraints and partial observability. In contrast, our approach employs a factored-POMDP setting, explicitly enabling the learning of the transition graph. In biological scenarios, our technique is reminiscent of "trajectory replay" in rodent planning, where neural patterns associated with past experiences are replayed in both forward and reverse directions (Ólafsdóttir et al., 2018). Thus, our method holds value for applications where offline trajectories can be leveraged.

In online settings, our hindsight encoder and causal transition model, initially trained on offline trajectories, can support model rollouts for action planning using strategies such as Model Predictive Control (Mayne et al., 2000) or Cross-Entropy Method (Botev et al., 2013). Specifically, given the transition data up to the current time step $t$, the hidden state at $t-n$ can be encoded using the forward observations and actions from $t-n+1$ to $t$. The learned transition model is then applied recursively $n+1$ times to predict the states at $t+1$. Notably, in the case of a single hidden factor, only one-step future information suffices to identify the discrete value of the hidden factor, up to an invertible linear transformation. In the online prediction of such cases, the hidden factor can be encoded one step backward, and the transition model can be applied twice to predict the next states at $t+1$. This model-based planning can be combined with a model-free agent that leverages transitions collected through planned actions to train both the actor and critic, thereby enhancing sample efficiency (Nagabandi et al., 2018; Du et al., 2020). Furthermore, the inferred transition graph can be employed to extract action-relevant states for policy learning, as the original state space transitions are often dense and prone to spurious correlations in the learned policy (Wang et al., 2022).

Our work has the following limitations. First, it requires an independent factorization of state variables in a factored-POMDP form. Integrating our framework with methods that embed high-dimensional partial observations of more general POMDPs into low-dimensional, disentangled state representations (Hafner et al., 2019; 2023), specifically in a factored-POMDP form, would be highly beneficial (Schölkopf et al., 2021; Liu et al., 2023). Such an approach could not only yield more accurate hidden representations than typical past-based methods but also facilitate the identification of causal transition connections. Second, we assume that the actions taken in the offline trajectory data are sufficiently diverse to explore the entire state-action space (even if state is not fully observed), enabling full system identification. The uniformly distributed collection policy used in our modulo environment satisfies this condition. However, in more complex environments where the state-action space is not fully explored (even if fully observed), the unexplored regions of the space will manifest as epistemic noise in the hidden states, resulting in biases in the identified hidden representations. In such cases, policy learning for active transition data collection becomes necessary (Seitzer et al., 2021; Wang et al., 2022; Jarrett et al., 2023).

In our formulation, we identified deterministic hidden components of factored state transitions, and, using the Reparametrization Lemma, isolated stochastic effects as unobserved exogenous noise per factor. Future work could refine our framework by also inferring the exogenous noise at each time step through dedicated noise encoders, following the identification of deterministic hidden factors (see Appendix B.5 for a detailed discussion). While our DVAE 1-step Hindsight Encoder was sufficient for a single hidden factor, extending it

to scenarios with multiple cascaded hidden factors, with only the last hidden factor influencing an observable factor, may require additional future information for effective latent identification. Moreover, expanding this approach to continuous state-action spaces would link our work to DVAE research on latent dynamics in stochastic-driven dynamical systems (Girin et al., 2020). Our framework has limited applicability in the real world, since we require a factored-POMDP, fixed initial conditions for the hidden factors, restrictions on scalability to multiple hidden factors, and an offline dataset that covers all state-action transitions. Addressing these areas would support further scaling and generalization of the framework.

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

# A    DVAE for Factored-POMDP

## A.1    Log-likelihood decomposition

The detailed derivation from Eq. 4 to Eq. 5 is provided as follows:

$$
\mathbb{E}_{p(o_{1:T}|a_{1:T})}\left[\log p_\theta(o_{1:T}|a_{1:T})\right]
$$

$$
=\mathbb{E}_{p(o_{1:T}|a_{1:T})}\left[\mathbb{E}_{q_\phi(h_{1:T}|o_{1:T},a_{1:T})}\left[\log p_\theta(o_{1:T}|a_{1:T})\right]\right] \tag{14}
$$

$$
=\mathbb{E}_{p(o_{1:T}|a_{1:T})}\left[\mathbb{E}_{q_\phi(h_{1:T}|o_{1:T},a_{1:T})}\left[\log \frac{p_\theta(o_{1:T},h_{1:T}|a_{1:T})}{p_\theta(h_{1:T}|o_{1:T},a_{1:T})}\right]\right] \tag{15}
$$

$$
=\mathbb{E}_{p(o_{1:T}|a_{1:T})}\left[\mathbb{E}_{q_\phi(h_{1:T}|o_{1:T},a_{1:T})}\left[\log \frac{p_\theta(o_{1:T},h_{1:T}|a_{1:T})}{q_\phi(h_{1:T}|o_{1:T},a_{1:T})}\frac{q_\phi(h_{1:T}|o_{1:T},a_{1:T})}{p_\theta(h_{1:T}|o_{1:T},a_{1:T})}\right]\right] \tag{16}
$$

$$
=\mathbb{E}_{p(o_{1:T}|a_{1:T})}\left[\mathbb{E}_{q_\phi(h_{1:T}|o_{1:T},a_{1:T})}\left[\log \frac{p_\theta(o_{1:T},h_{1:T}|a_{1:T})}{q_\phi(h_{1:T}|o_{1:T},a_{1:T})}\right]\right.
$$

$$
\left.+\ \mathbb{E}_{q_\phi(h_{1:T}|o_{1:T},a_{1:T})}\left[\log \frac{q_\phi(h_{1:T}|o_{1:T},a_{1:T})}{p_\theta(h_{1:T}|o_{1:T},a_{1:T})}\right]\right] \tag{17}
$$

$$
=\mathbb{E}_{p(o_{1:T}|a_{1:T})}\Big[\underbrace{\mathbb{E}_{q_\phi(h_{1:T}|o_{1:T},a_{1:T})}\left[\log p_\theta\left(o_{1:T},h_{1:T}|a_{1:T}\right)-\log q_\phi\left(h_{1:T}|o_{1:T},a_{1:T}\right)\right]}_{\ell_{\mathrm{VLB}}(\theta,\phi;o_{1:T},a_{1:T})}
$$

$$
+\ D_{\mathrm{KL}}(q_\phi(h_{1:T}|o_{1:T},a_{1:T})\parallel p_\theta(h_{1:T}|o_{1:T},a_{1:T}))\Big] \tag{18}
$$

## A.2    VLB of DVAE-based framework for learning the hidden encoder and transition dynamics of a factored-POMDP

**Generative model (transition model).** The generative model for the entire state sequence in the Eq. 6 can be factorized as:

$$
p_\theta(o_{1:T},h_{1:T}|a_{1:T})=\prod_{t=0}^{T-1}p_\theta(o_{t+1},h_{t+1}|o_{1:t},h_{1:t},a_{1:T})
$$

$$
=\prod_{t=0}^{T-1}p_{\theta_o}\left(o_{t+1}|o_{1:t},h_{1:t+1},a_{1:T}\right)p_{\theta_h}\left(h_{t+1}|o_{1:t},h_{1:t},a_{1:T}\right)
$$

$$
=\prod_{t=0}^{T-1}p_{\theta_o}\left(o_{t+1}|o_t,h_t,a_t\right)p_{\theta_h}\left(h_{t+1}|o_t,h_t,a_t\right) \tag{19}
$$

where each term in the product is simplified using d-separation in the unrolled transition graph from $t=1$ to $T$ (see Fig. 1c). Here, $\theta=\theta_o\cup\theta_h$ represents the parameters of the generative model. Note that the observation likelihood $p_{\theta_o}\left(o_{t+1}|o_t,h_t,a_t\right)$ and the hidden prior $p_{\theta_h}\left(h_{t+1}|o_t,h_t,a_t\right)$ in the generative model correspond to the transition models of the observed and hidden states, respectively.

**Inference model (hidden encoder).** Similarly, we factorize the posterior distribution of the generative model as follows:

$$
p_\theta(h_{1:T}|o_{1:T},a_{1:T})=\prod_{t=0}^{T-1}p_\theta(h_{t+1}|h_{1:t},o_{1:T},a_{1:T})
$$

$$
=\prod_{t=0}^{T-1}p_\theta(h_{t+1}|h_t,o_{t:T},a_{t:T}) \tag{20}
$$

We consider that the inference model, parameterized by $\phi$, captures the exact factorized structure of the posterior distribution in Eq. 20:

$$
q_\phi(h_{1:T}|o_{1:T},a_{1:T})=\prod_{t=0}^{T-1}q_\phi(h_{t+1}|h_t,o_{t:T},a_{t:T}) \tag{21}
$$

Specifically, the hidden encoder $q_\phi(h_t|h_{t-1}, o_{t-1:T}, a_{t-1:T})$ combines information from the Markovian past, through $h_{t-1}$, $o_{t-1}$ and $a_{t-1}$, with information from the present and future observations $o_{t:T}$ and actions $a_{t:T}$ to encode the current hidden state $h_t$. It is important to note that we assume Markovianity for the forward transitions but not for the backward transitions. Consequently, the hidden encoder depends on all future information, rather than just the immediate next-step information used in the hindsight-based encoder by Jarrett et al. (2023).

**Variational Lower Bound.** By substituting the decomposed forms of both the generative model from Eq. 19 and the inference model from Eq. 21 into the general form of VLB defined in Eq. 6, we obtain:

$$
\ell_{\text{VLB}}\left(\theta, \phi; o_{1:T}, a_{1:T}\right)
$$

$$
=\mathbb{E}_{q_\phi(h_{1:T}|o_{1:T}, a_{1:T})}\left[\log\prod_{t=0}^{T-1} p_{\theta_o}\left(o_{t+1}|o_t, h_t, a_t\right) p_{\theta_h}\left(h_{t+1}|o_t, h_t, a_t\right) - \log\prod_{t=0}^{T-1} q_\phi(h_{t+1}|h_t, o_{t:T}, a_{t:T})\right]
$$

$$
=\sum_{t=0}^{T-1}\left[\mathbb{E}_{q_\phi(h_{t+1:T}|h_{1:t}, o_{1:T}, a_{1:T})q_\phi(h_{1:t}|o_{1:T}, a_{1:T})}\left[\log p_{\theta_o}\left(o_{t+1}|o_t, h_t, a_t\right)\right]\right.
$$

$$
\left. -\mathbb{E}_{q_\phi(h_{t+2:T}|h_{1:t+1}, o_{1:T}, a_{1:T})q_\phi(h_{1:t+1}|o_{1:T}, a_{1:T})}\left[\log\left(q_\phi(h_{t+1}|h_t, o_{t:T}, a_{t:T})/p_{\theta_h}\left(h_{t+1}|o_t, h_t, a_t\right)\right)\right]\right]
$$

$$
=\sum_{t=0}^{T-1}\left[\mathbb{E}_{q_\phi(h_{1:t}|o_{1:T}, a_{1:T})}\left[\log p_{\theta_o}\left(o_{t+1}|o_t, h_t, a_t\right)\right]\right.
$$

$$
\left. -\mathbb{E}_{q_\phi(h_{1:t+1}|o_{1:T}, a_{1:T})}\left[\log\left(q_\phi(h_{t+1}|h_t, o_{t:T}, a_{t:T})/p_{\theta_h}\left(h_{t+1}|o_t, h_t, a_t\right)\right)\right]\right]
$$

$$
=\sum_{t=0}^{T-1}\left[\mathbb{E}_{q_\phi(h_{1:t}|o_{1:T}, a_{1:T})}\left[\log p_{\theta_o}\left(o_{t+1}|o_t, h_t, a_t\right)\right]\right.
$$

$$
\left. -\mathbb{E}_{q_\phi(h_{1:t}|o_{1:T}, a_{1:T})}\mathbb{E}_{q_\phi(h_{t+1}|h_t, o_{t:T}, a_{t:T})}\left[\log\left(q_\phi(h_{t+1}|h_t, o_{t:T}, a_{t:T})/p_{\theta_h}\left(h_{t+1}|o_t, h_t, a_t\right)\right)\right]\right]
$$

$$
=\sum_{t=0}^{T-1}\mathbb{E}_{q_\phi(h_{1:t}|o_{1:T}, a_{1:T})}\left[\log p_{\theta_o}\left(o_{t+1}|o_t, h_t, a_t\right)\right.
$$

$$
\left. -D_{\text{KL}}\left(q_\phi\left(h_{t+1}|h_t, o_{t:T}, a_{t:T}\right)\parallel p_{\theta_h}\left(h_{t+1}|o_t, h_t, a_t\right)\right)\right] \tag{22}
$$

By using the factorization in Eq. 21, the expectation in the above VLB can be expressed as a cascade of expectations over conditional distributions of individual hidden states at different time steps:

$$
\mathbb{E}_{q_\phi(h_{1:t}|o_{1:T}, a_{2:T})}\left[f\left(h_t\right)\right] =\mathbb{E}_{q_\phi(h_1|o_{1:T}, a_{1:T})}\big[
$$
$$
\mathbb{E}_{q_\phi(h_2|h_1, o_{1:T}, a_{1:T})}\big[
$$
$$
\mathbb{E}_{q_\phi(h_3|h_2, o_{2:T}, a_{2:T})}\big[\cdots
$$
$$
\mathbb{E}_{q_\phi(h_t|h_{t-1}, o_{t-1:T}, a_{t-1:T})}\left[f\left(h_t\right)\right]\cdots\big]\big]\big] \tag{23}
$$

Here, $f\left(h_t\right)$ represents an arbitrary function of $h_t$. Each intractable expectation in this sequence can be approximated using a Monte Carlo estimate. This involves iteratively sampling from $q_\phi(h_\tau|h_{\tau-1}, o_{\tau-1:T}, a_{\tau-1:T})$ for $\tau = 1$ to $t$, employing the same reparameterization trick used in standard VAEs (Maddison et al., 2016; Jang et al., 2016; Kingma & Welling, 2019). Additionally, the VLB in Eq. 22, which is defined for a single data sequence, can be extended by averaging over a mini-batch of training data sequences, thereby approximating the expected VLB with respect to the true data distribution.

Furthermore, by expressing $o_t = (o_t^1, \ldots, o_t^{d_O})$, $h_t = (h_t^1, \ldots, o_t^{d_H})$ and $s_t = (o_t, h_t)$ and using the assumption of factorized transition dynamics, we have:

$$p_{\theta_o}(o_{t+1}|o_t, h_t, a_t) = \prod_{j=1}^{d_O} p_{\theta_h}(o_{t+1}^j|s_t, a_t), \tag{24}$$

$$p_{\theta_h}(h_{t+1}|o_t, h_t, a_t) = \prod_{j=1}^{d_H} p_{\theta_h}(h_{t+1}^j|s_t, a_t), \tag{25}$$

$$q_\phi(h_{t+1}|h_t, o_{t:T}, a_{t:T}) = \prod_{j=1}^{d_H} q_\phi(h_{t+1}^j|h_t, o_{t:T}, a_{t:T}) \tag{26}$$

Eq. 8 is obtained by substituting the above expressions into Eq. 22.

### A.3 Conditional mutual information

Starting from the definition of conditional mutual information, we have:

$$I(s_t^i; s_{t+1}^j|s_t \backslash s_t^i, a_t) = \mathbb{E}_{p(s_t, a_t, s_{t+1}^j)} \left[ \log \frac{p(s_t^i, s_{t+1}^j|s_t \backslash s_t^i, a_t)}{p(s_t^i|s_t \backslash s_t^i, a_t)p(s_{t+1}^j|s_t \backslash s_t^i, a_t)} \right] \tag{27}$$

$$= \mathbb{E}_{p(s_t, a_t, s_{t+1}^j)} \left[ \log \frac{p(s_{t+1}^j|s_t, a_t)p(s_t^i|s_t \backslash s_t^i, a_t)}{p(s_t^i|s_t \backslash s_t^i, a_t)p(s_{t+1}^j|s_t \backslash s_t^i, a_t)} \right] \tag{28}$$

$$= \mathbb{E}_{p(s_t, a_t, s_{t+1}^j)} \left[ \log \frac{p(s_{t+1}^j|s_t, a_t)}{p(s_{t+1}^j|s_t \backslash s_t^i, a_t)} \right] \tag{29}$$

$$= \mathbb{E}_{p(s_t, a_t)} \left[ \mathbb{E}_{p(s_{t+1}^j|s_t, a_t)} \left[ \log \frac{p(s_{t+1}^j|s_t, a_t)}{p(s_{t+1}^j|s_t \backslash s_t^i, a_t)} \right] \right] \tag{30}$$

$$= \mathbb{E}_{p(s_t, a_t)} \left[ D_{\mathrm{KL}}(p(h_{t+1}^j|s_t, a_t) \,\|\, p(h_{t+1}^j|s_t \backslash s_t^i, a_t)) \right] \tag{31}$$

where Eqs. 29 and 31 correspond to Eqs. 9 and 10, respectively.

## A.4 Algorithm Details

---

**Algorithm 1** Causal Dynamics Learning with Hindsight

---

**Input**: Initial hidden encoder $q_\phi$, initial transition models $p_{\theta_o}$ and $p_{\theta_h}$, initial reward predictor $R_\psi$, and replay buffer $\mathcal{D}$ containing pre-collected data.

**Parameters**: Learning rate $\alpha > 0$, CMI threshold $\delta > 0$, training steps $M$, CMI eval. period $N$.

**Output**: Converged hidden encoder $q_{\phi^*}$, transition models $p_{\theta_o^*}$, $p_{\theta_h^*}$, and graph $\mathcal{G}^*$, reward predictor $R_{\psi^*}$.

1: **for** $k = 1$ to $M$ training steps **do**
2:      Update $\mathcal{D}$ and randomly sample a minibatch of $m$ episodes $\{o_{1:T}^{(e)}, a_{1:T}^{(e)}, \tau_{1:T}^{(e)}, r_{1:T}^{(e)}\}_{e=1}^m$.
3:      Compute the mean objective $\mathcal{L}_{\text{obj}}(\theta, \phi, \overline{\phi}, \psi; o_{1:T}^{(1:m)}, a_{1:T}^{(1:m)}, \tau_{1:T}^{(1:m)}, r_{1:T}^{(1:m)})$ using Eq. 11.
4:      Update the model parameters:

$$[\theta_o, \theta_h, \phi, \psi] \leftarrow [\theta_o, \theta_h, \phi, \psi] + \alpha \nabla \mathcal{L}_{\text{obj}}(\theta, \phi, \overline{\phi}, \psi; o_{1:T}^{(1:m)}, a_{1:T}^{(1:m)}, \tau_{1:T}^{(1:m)}, r_{1:T}^{(1:m)})$$
$$\overline{\phi} \leftarrow \phi$$

5:      **if** $k \bmod N = 0$ **then**
6:          Evaluate $\text{CMI}^{i,j}$ using Eqs. 9 and 10, and update with an exponential moving average.
7:          Binarize $\text{CMI}^{i,j}$ to construct $\mathcal{G}$ by checking if $\text{CMI}^{i,j} \geq \delta$.
8:      **end if**
9: **end for**

---

## A.5 Neural Network-Based Parameterization

The hidden encoder $q_\phi(h_t | h_{t-1}, o_{t-1:T}, a_{t-1:T})$ is implemented using a backward RNN to capture current and future dependencies, and an MLP to model Markovian past dependencies. A combiner function (CF) is then employed to merge the outputs of the MLP and the RNN (its internal state) to produce parameters (e.g., logits) of the distribution of the current hidden state:

$$\overleftarrow{g}_t = \text{RNN}_{\phi_{\overleftarrow{g}}}(\overleftarrow{g}_{t+1}, [o_t, a_t]), \tag{32}$$

$$e_t = \text{MLP}_{\phi_e}(h_{t-1}, o_{t-1}, a_{t-1}), \tag{33}$$

$$f_t = \text{CF}_{\phi_f}(e_t, \overleftarrow{g}_t), \tag{34}$$

$$q_\phi(h_t | h_{t-1}, o_{t-1:T}, a_{t-1:T}) = \text{Dist}(h_t; f_t) \tag{35}$$

where $\text{CF}_{\phi_f}$ is a feedforward combining network parameterized by $\phi_f$. Thus, the parameters of the hidden encoder are $\phi = \phi_{\overleftarrow{g}} \cup \phi_e \cup \phi_f$.

The transition model for the observed states $p_{\theta_o}(o_{t+1} | o_t, h_t, a_t)$ and the hidden states $p_{\theta_h}(h_{t+1} | o_t, h_t, a_t)$ are implemented using factor-wise Masked MLPs (MMLPs) following Wang et al. (2022):

$$m_t = \text{MMLP}_{\theta_o}(o_t, h_t, a_t), \tag{36}$$

$$p_{\theta_o}(o_{t+1} | o_t, h_t, a_t) = \text{Dist}(o_t; m_t), \tag{37}$$

$$n_t = \text{MMLP}_{\theta_h}(o_t, h_t, a_t), \tag{38}$$

$$p_{\theta_h}(h_{t+1} | o_t, h_t, a_t) = \text{Dist}(h_t; n_t) \tag{39}$$

where $m_t$ and $n_t$ are the outputs of the masked MLPs parameterized by $\theta_o$ and $\theta_h$, respectively. The distributions $\text{Dist}(o_t; m_t)$ and $\text{Dist}(h_t; n_t)$ represent the probability distributions of $o_{t+1}$ and $h_{t+1}$ parameterized by $m_t$ and $n_t$.

Specifically, each factor-wise MMLP models the factorized transition probability of an individual state factor. It accepts the complete set of current state factors and actions $\{s_t, a_t\}$ as inputs and outputs the distribution for a specific state factor $j$ at the next time step, $p(s_{t+1}^j | s_t, a_t)$. We employ $d_S$ number of distinct MMLPs, one for each state factor, to represent the factorized transition dynamics comprehensively. Moreover, the influence of a particular state or action factor on the output of the MMLP can be selectively removed

by masking the representation of that factor—setting it to negative infinity—and subsequently taking the maximum across the representations of all input factors within the MMLP.

The architecture of the DVAE model is illustrated in Fig. 5.

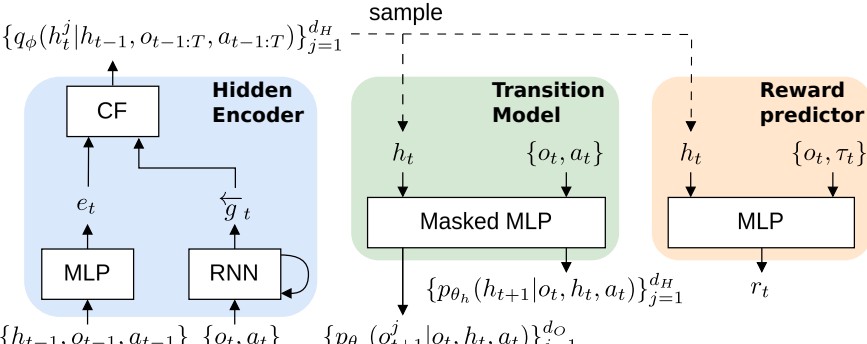

Figure 5: Model architecture illustrating the computational graph for encoding, sampling and prediction processes.

# B    Additional results

## B.1    Zoomed-in training dynamics

Fig. 6 zooms into the x-axis of Fig. 4 to be able to distinguish the early loss profiles of the various encoders that overlap after convergence.

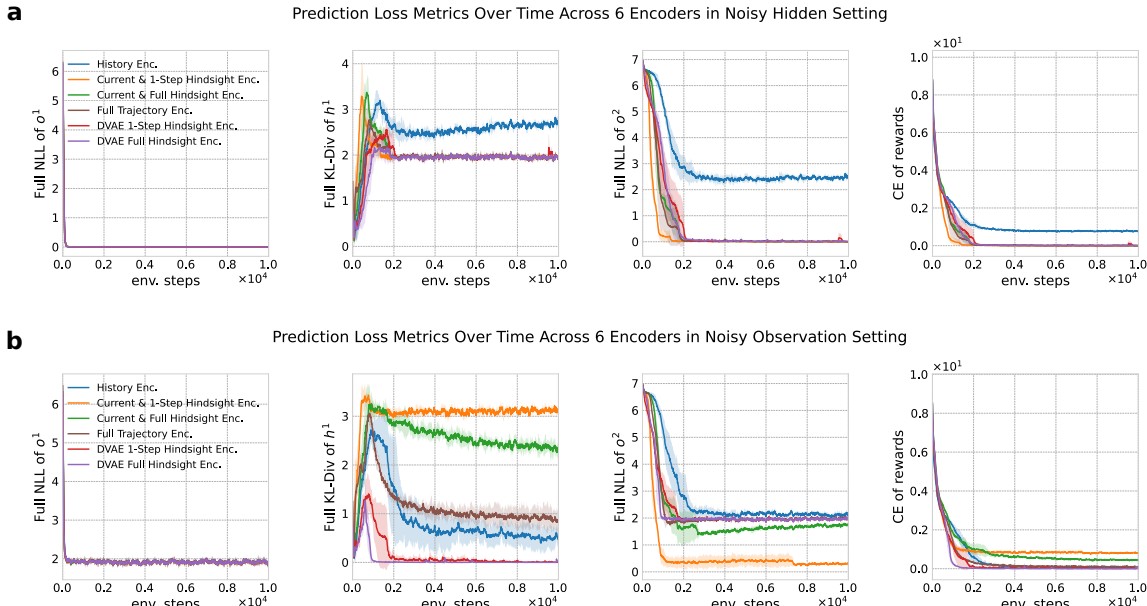

Figure 6: Zooming in on the x-axis of training dynamics in Fig. 4 to highlight the initial transient loss profile.

## B.2    Learning transition graphs with varying structures

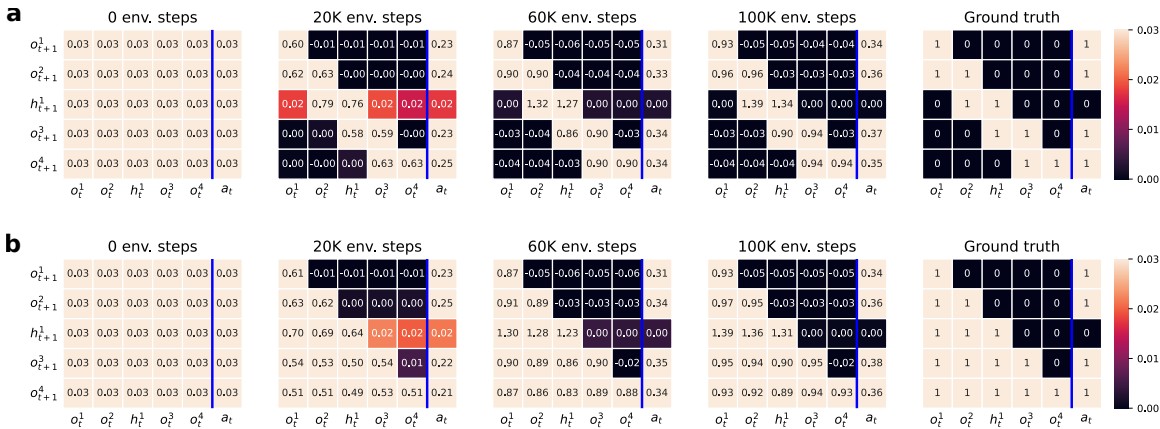

Figure 7: Evolution of CMI matrices for the **(a)** chain and **(b)** full graph structures. The ground truth graphs are shown on the far right.

We evaluate our DVAE 1-step Hindsight Encoder on different transition graph structures using a modulo environment with parameters $d_S = 5$, $l = 4$ and a single hidden state. All other aspects of the environment setup remain unchanged from those described in Section 4. Fig. 7 shows the evolution of the CMI matrix

during training in the setting of noisy observations for chain and full structured (lower-triangular adjacency matrix) transition graphs, respectively. The CMI matrix initially has all elements set to the predefined threshold $\delta$ and gradually decreases for unconnected factor pairs in the transition while increasing for connected factor pairs. The final binary matrix, obtained by applying the threshold to binarize the CMI matrix, converges to the ground truth adjacency matrix.

## B.3 Scalability to Larger State Spaces

We evaluate scalability on larger state spaces using a modulo environment characterized by a chain-structured transition graph with parameters $d_S = 5$, $l = 5$ and a single hidden state. The distributions for noise and initial states remain consistent with those described in Section 4. We compare six baseline methods under a noisy observation setting with a single hidden factor. Training and evaluation performances are presented in Fig. 8 and Table 2, respectively. The convergence of losses for the hidden state $h^1$ and reward to non-zero values suggests that learning does not achieve global optimality, after running each baseline for $\sim 15$ hours on a single A100 GPU, likely due to the inherent difficulties associated with gradient-based optimization in discrete transition dynamics (Niculae et al., 2023). Nevertheless, our DVAE 1-step hindsight encoder consistently achieves the lowest prediction losses for both the hidden state and reward (see top-right and bottom-right panels in Fig. 8), as well as superior evaluation accuracy across most metrics (second-to-last row of Table 2). Notably, the high prediction accuracy for observation $o^3$ achieved by the current and 1-step hindsight encoders can be attributed to a pathological solution wherein the encoder infers $h_t^1$ directly as a copy of its child observation $o_{t+1}^3$, simplifying the transition model for $o_{t+1}^3$ to an identity mapping. This solution trades off the prediction accuracy of the hidden state $h^1$ and reward to achieve perfect prediction for the noisy observation $o^3$.

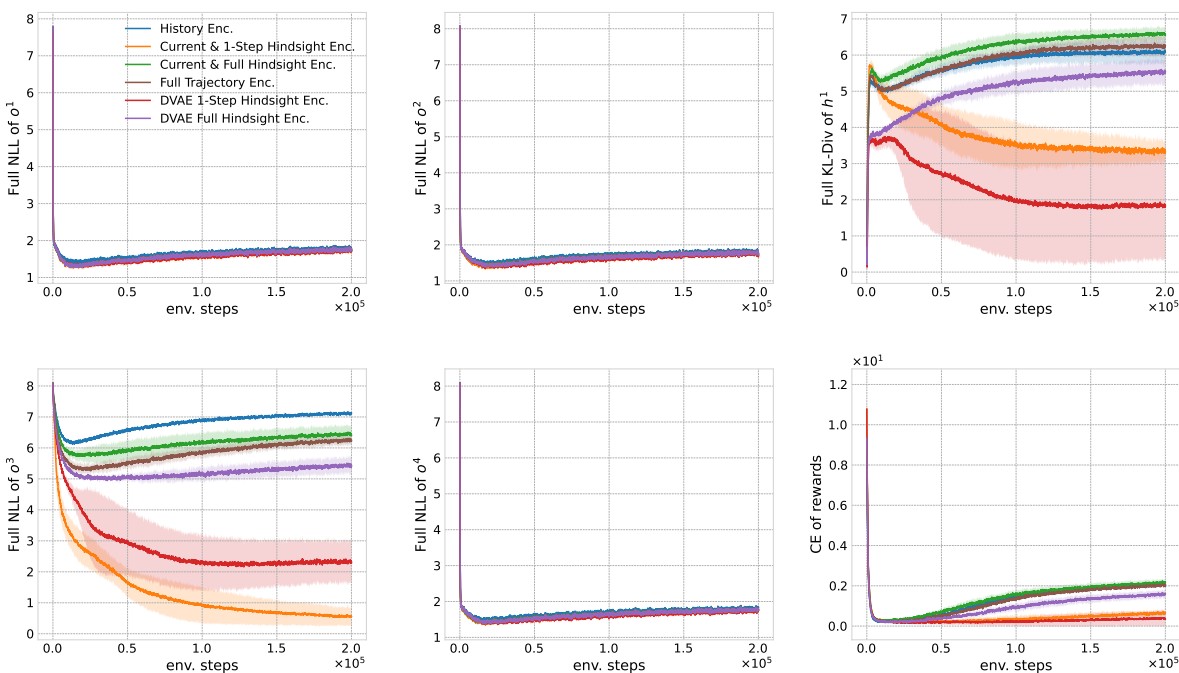

Figure 8: Training performance profiles for six encoder types under a noisy observation setting with one hidden factor. The environment utilizes a chain-structured transition graph ($d_S = 5$, $l = 5$) comprising five factors: two observable states ($o^1$, $o^2$), a central hidden state ($h^1$), and two additional observable states ($o^3$, $o^4$).

| # Past | # Future | Graph | $h^1$ Decoding | $o^1, o^2, o^3, o^4$ Prediction | | | | $h^1$ Prediction | Reward Prediction |
|---|---|---|---|---|---|---|---|---|---|
| | | | | Evaluation Accuracy in Noisy Observation Setting | | | | | |
| **History-Based Encoder** | | | | | | | | | |
| all | 0 | $0.833_{(0.000)}$ | $0.329_{(0.013)}$ | $0.897_{(0.023)}$ | $0.896_{(0.006)}$ | $0.371_{(0.010)}$ | $0.897_{(0.012)}$ | $0.401_{(0.040)}$ | $0.692_{(0.020)}$ |
| **Current and Hindsight-Based Encoder** | | | | | | | | | |
| 0 | 1 | $0.800_{(0.047)}$ | $0.866_{(0.024)}$ | $0.896_{(0.023)}$ | $0.895_{(0.005)}$ | $0.960_{(0.025)}$ | $0.897_{(0.012)}$ | $0.750_{(0.043)}$ | $0.897_{(0.029)}$ |
| 0 | all | $0.844_{(0.016)}$ | $0.368_{(0.024)}$ | $0.897_{(0.023)}$ | $0.895_{(0.006)}$ | $0.434_{(0.032)}$ | $0.897_{(0.012)}$ | $0.333_{(0.040)}$ | $0.754_{(0.015)}$ |
| **Full Trajectory-Based Encoder** | | | | | | | | | |
| all | all | $0.867_{(0.000)}$ | $0.419_{(0.023)}$ | $0.896_{(0.023)}$ | $0.896_{(0.006)}$ | $0.458_{(0.008)}$ | $0.897_{(0.012)}$ | $0.398_{(0.049)}$ | $0.779_{(0.018)}$ |
| **DVAE-Based Hindsight Encoder** | | | | | | | | | |
| all | 1 | $0.911_{(0.063)}$ | $0.895_{(0.095)}$ | $0.897_{(0.023)}$ | $0.896_{(0.006)}$ | $0.846_{(0.084)}$ | $0.896_{(0.012)}$ | $0.849_{(0.150)}$ | $0.927_{(0.062)}$ |
| all | all | $0.867_{(0.000)}$ | $0.516_{(0.035)}$ | $0.896_{(0.023)}$ | $0.896_{(0.006)}$ | $0.528_{(0.039)}$ | $0.897_{(0.011)}$ | $0.459_{(0.055)}$ | $0.789_{(0.024)}$ |

Table 2: Evaluation accuracy metrics across six encoder types under the environment configuration described in Fig. 8. The highest accuracy values for each metric across encoders are highlighted in beige.

## B.4   Scalability to Two Hidden Factors

To assess scalability in scenarios involving two hidden states, we again utilize the modulo environment with a chain-structured transition graph parameterized by $d_S = 5$ and $l = 3$. We compare the same six baseline methods under noisy observation conditions with two non-consecutive hidden factors $h^1, h^2$ located at positions 2 and 4, and observable factors $o^1, o^2, o^3$ at positions 1, 3 and 5, in the chain. Training and evaluation results are shown in Fig. 9 and Table 3, respectively. Despite the convergence of losses for the two hidden states $h^1, h^2$ and reward to sub-optimal values across all models, our DVAE 1-step hindsight encoder achieves consistently superior results, demonstrating the lowest prediction losses for both hidden states and reward (refer to the top-middle, bottom-left, and bottom-right panels in Fig. 9), as well as the highest evaluation accuracy in most metrics (second-to-last row in Table 3). Similar to the previous scenario, the exceptional prediction accuracy of the noisy observation $o^3$ by the 1-step hindsight encoder stems from the encoder's tendency to adopt a copying strategy, resulting in a simplified, identity-based transition model.

| # Past | # Future | Graph | $h^1, h^2$ Decoding | $o^1, o^2, o^3$ Prediction | | | $h^1, h^2$ Prediction | Reward Prediction |
|---|---|---|---|---|---|---|---|---|
| | | | | Evaluation Accuracy in Noisy Observation Setting | | | | |
| **History-Based Encoder** | | | | | | | | |
| all | 0 | $0.811_{(0.134)}$ | $0.817_{(0.087)}, 0.864_{(0.160)}$ | $0.901_{(0.008)}$ | $0.771_{(0.048)}$ | $0.820_{(0.095)}$ | $0.725_{(0.148)}, 0.861_{(0.106)}$ | $0.819_{(0.087)}$ |
| **Current and Hindsight-Based Encoder** | | | | | | | | |
| 0 | 1 | $0.844_{(0.113)}$ | $0.886_{(0.008)}, 0.891_{(0.015)}$ | $0.901_{(0.008)}$ | $0.991_{(0.006)}$ | $0.995_{(0.006)}$ | $0.785_{(0.028)}, 0.791_{(0.041)}$ | $0.835_{(0.006)}$ |
| 0 | all | $0.856_{(0.096)}$ | $0.576_{(0.037)}, 0.570_{(0.053)}$ | $0.901_{(0.008)}$ | $0.605_{(0.027)}$ | $0.603_{(0.058)}$ | $0.470_{(0.012)}, 0.438_{(0.028)}$ | $0.556_{(0.051)}$ |
| **Full Trajectory-Based Encoder** | | | | | | | | |
| all | all | $0.789_{(0.079)}$ | $0.592_{(0.073)}, 0.539_{(0.031)}$ | $0.901_{(0.008)}$ | $0.592_{(0.069)}$ | $0.546_{(0.021)}$ | $0.450_{(0.042)}, 0.483_{(0.049)}$ | $0.536_{(0.064)}$ |
| **DVAE-Based Hindsight Encoder** | | | | | | | | |
| all | 1 | $0.933_{(0.094)}$ | $0.926_{(0.083)}, 0.916_{(0.088)}$ | $0.901_{(0.008)}$ | $0.860_{(0.048)}$ | $0.858_{(0.051)}$ | $0.911_{(0.104)}, 0.902_{(0.104)}$ | $0.909_{(0.095)}$ |
| all | all | $0.733_{(0.000)}$ | $0.864_{(0.193)}, 0.842_{(0.223)}$ | $0.901_{(0.008)}$ | $0.791_{(0.161)}$ | $0.779_{(0.183)}$ | $0.806_{(0.274)}, 0.831_{(0.239)}$ | $0.843_{(0.222)}$ |

Table 3: Evaluation accuracy metrics across six encoder types under the environment configuration described in Fig. 9. The highest accuracy values for each metric across encoders are highlighted in beige.

## B.5   Inference over hidden and noise factors

In the noisy observation setting, with a single hidden factor $h_t$, we can introduce $d_O$ additional noise encoders $q_{\phi_\epsilon}(\epsilon_t^i | o_t, h_t, a_t)$, one for each noise variable $\epsilon_t^i$. Each noise encoder is conditioned on the latent state $h_t$, inferred by our DVAE 1-step hindsight encoder $q_\phi(h_t | h_{t-1}, o_{t-1:t+1}, a_{t-1:t+1})$, as well as on the current observation $o_t$ and action $a_t$. To ensure exogeneity of the noise factors $\epsilon_t^1, \ldots, \epsilon_t^{d_O}$, we require each of them to be independent of the state-action pair $o_t, h_t, a_t$. Analogous to the CMI for the next hidden state in Eq. 10, we measure the factor-wise mutual information between each noise factor $\epsilon_t^i$ and the state-action pair

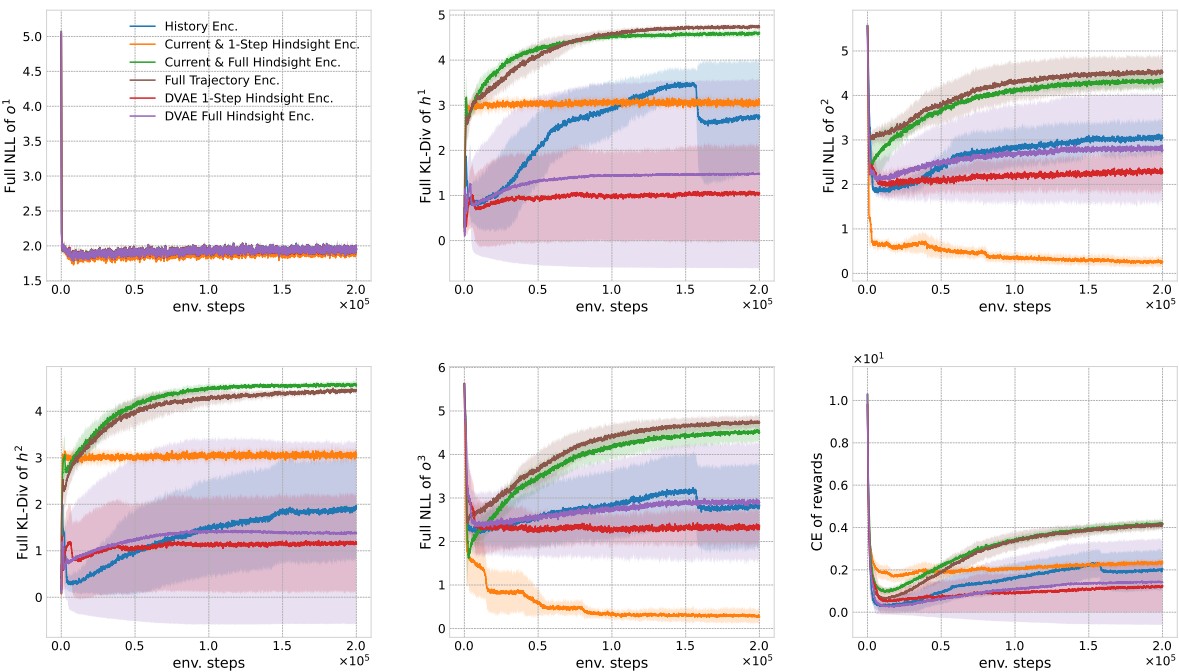

Figure 9: Training performance profiles for six encoder types under a noisy observation setting with two hidden factors. The environment utilizes a chain-structured transition graph ($d_S = 5$, $l = 3$) comprising five factors: an initial observable state ($o^1$), the first hidden state ($h^1$), an intermediate observable state ($o^2$), the second hidden state ($h^2$), and a final observable state ($o^3$).

$o_t, h_t, a_t$ as

$$I_{\phi_\epsilon}(\epsilon_t^i; o_t, h_t, a_t) = \mathbb{E}_{o_t, h_t, a_t \sim \mathcal{D}, q_\phi}[D_{\mathrm{KL}}(q_{\phi_\epsilon}(\epsilon_t^i | o_t, h_t, a_t) \parallel q_{\phi_\epsilon}(\epsilon_t^i))] \tag{40}$$

By driving the sum of these factor-wise mutual information terms across all noise factors to zero,

$$\min_{\phi_\epsilon} \sum_{i=1}^{d_O} I_{\phi_\epsilon}(\epsilon_t^i; o_t, h_t, a_t) \tag{41}$$

we enforce independence between the noise factors and the state-action pair, thereby preserving the exogeneity of the noise.

However, if multiple current hidden factors collide on a conditioned future observation factor in the unrolled transition graph, the path between these factors becomes unblocked. In this situation, the DVAE-based encoder can no longer unequivocally disentangle these hidden factors, let alone separate them from the noise.

