# OpenReview forum: "Dynamical-VAE-based Hindsight to Learn the Causal Dynamics of Factored-POMDPs"
_TMLR — Rejected by TMLR_

### Review · Reviewer_uoD6 · 2025-03-03

**Summary Of Contributions:**

This paper tackles the problem of learning the underlying Markovian dynamics in a partially observable Markov decision process (POMDP), focusing on both estimating the dynamics function and identifying the unobservable portion of the state. The authors present a Dynamical Variational Auto-Encoder (DVAE) that integrates past, current, and multi-step future information to infer hidden state factors. Experiments on a one-hidden-factor factored-POMDP demonstrate that this approach successfully uncovers the hidden state and effectively learns the underlying causal model.

**Audience:**

Yes

**Broader Impact Concerns:**

Although the authors do not include a dedicated **Broader Impact Statement** in the paper, the discussion provided in **Section 5** partially addresses the broader implications of their work. Specifically, they briefly touch on the limitation and considerations that arise when leveraging their method in real-world systems: multiple restrictive assumptions should be expanded to address the more complex challenges in the real world, like the independent observations and hidden states, the exploration and coverage of random policies, etc.

**Claims And Evidence:**

Yes

**Requested Changes:**

### Requests for Changes

1. **Clarify Offline Data Collection Procedure and Policy**
   - Elaborate on how the offline data was generated (e.g., in Section 3.1 or 3.2). Specify whether the policy was random or exploratory, and explain why actions appear not to depend directly on states in the causal graph.

2. **State Initialization Details**
   - Clearly state any assumptions about the initial state (hidden or observed) and how they factor into the learning process. Indicating whether the initial state is randomly sampled or fixed will clarify the applicability of the method.

3. **Generalization to More Complex Observation Models**
   - Discuss how the approach could be extended to settings where observations are nonlinear transformations of hidden states. Providing insight into whether the model can handle more realistic assumptions would enhance its practical relevance.

4. **Further Explanation of Additional Loss Terms**
   - Expand on the reasoning behind Equations (8) and (9), describing each loss component in detail (Full NLL, Masked NLL, Causal NLL, Masked KL-Div, Causal KL-Div). Explain their individual purposes and how the differences between “Full” and “Masked” losses reveal the underlying causal structure.

5. **Define \(\tau_t\) Explicitly**
   - Provide a precise definition of \(\tau_t\), clarifying whether it is an observable or hidden subset of the state. This information is crucial for understanding its role within the proposed framework.

**Strengths And Weaknesses:**

### Strengths
- The paper is clearly written, making it easy to follow and understand the core ideas.



### Weaknesses

1. **Data Collection and Policy Assumption**
   - The causal graph in Figure 1 does not include edges from state to action, implying that actions might be sampled from a random policy. However, while reading the corresponding section on the causal graph, it remains unclear how the authors collect offline data under this assumption, or what the actual behavior policy looks like. Although the authors mention this in Section 5 (Discussion), it appears too late in the paper. Including this clarification earlier—such as when introducing the method—would be more helpful.

2. **Initial State Assumption**
   - The paper does not specify assumptions or distributions regarding the initial hidden state or initial observation. This omission makes it difficult to determine how the proposed approach handles early-stage uncertainty in the trajectory.

3. **Strict Decomposition Assumption for the State Space**
   - The method assumes that the state space neatly decomposes into an observable component and an unobservable (hidden) component. However, in real-world scenarios, observations often arise from nonlinear transformations or mixtures of the hidden state, raising concerns about the generalizability of the proposed model given its relatively simple structure.

4. **Lack of Clarity in the Last Paragraph of Section/Page 5**
   - The shift from Equation (8) to Equation (9) and the rationale for multiple loss components (e.g., Full NLL, Masked NLL, Causal NLL, Masked KL-Divergence, Causal KL-Divergence) are not sufficiently detailed. Without referring to Wang et al. (2022), readers may not fully grasp how these different losses contribute to learning the causal graph or what the masking operation entails. Additional context is needed to bridge this gap.

5. **Ambiguity Regarding \(\tau_t\)**
   - It is unclear whether \(\tau_t\) represents part of the observation, part of the hidden state, or some other subset of state components. Moreover, the paper does not specify whether \(\tau_t\) is observable, making its role in the framework difficult to interpret.

6. **Limited Discussion of the Core Contribution**
   - The primary contribution is stated to be the incorporation of future information for dynamics learning and the hindsight part. However, the paper does not provide sufficient explanation or direct comparisons in the method section to demonstrate the intuitive benefits of using future information. A discussion on the identifiability of hidden states without future information would help clarify why future observations are important.

---

> ### Author Response · Authors · 2025-03-30
> **Responses to the reviewer - Part 1**
>
> We thank the reviewer for the comments and have addressed them below. We are happy to respond to any further feedback.
>
>
>
>  Comment 2.1:
> Clarify Offline Data Collection Procedure and Policy: Elaborate on how the offline data was generated (e.g., in Section 3.1 or 3.2). Specify whether the policy was random or exploratory, and explain why actions appear not to depend directly on states in the causal graph.
>
>  Response 2.1:
>
> We employed a random policy to collect offline trajectories. In our modulo environment setting, randomly chosen actions sufficiently explore the state-action space, ensuring that the offline dataset adequately covers most state-action transitions. In general, if actions are selected based on states rather than randomly, the action nodes in the unrolled transition graphs (Fig. 1(b) and (c)) would explicitly depend on states, whereas such dependence would not appear in the stationary transition graph shown in Fig. 1(a).
>
> In the revised manuscript, we have included the explanation above regarding the generation of offline data at the end of the Environment setup in Section 4, and we have also clarified the apparent independence of actions from states in the caption of Fig. 1.
>
> In the third paragraph of the Discussion section, we have also discussed the limitations of using a random policy for offline data collection and that an exploratory policy for active data collection would be more efficient in more complex environments.
>
>
>  Comment 2.2:
> State Initialization Details: Clearly state any assumptions about the initial state (hidden or observed) and how they factor into the learning process. Indicating whether the initial state is randomly sampled or fixed will clarify the applicability of the method.
>
>  Response 2.2:
>
> We assume that the initial hidden state remains fixed across episodes, while the initial observed states are randomly sampled in each episode.  Empirically, under noisy observation scenarios, we observed that this fixed initial hidden-state condition enables our encoder to disentangle the current hidden state from noise effectively. In contrast, when the initial hidden state is randomly sampled, all encoders fail to achieve such disentanglement and instead infer spurious correlations.
>
> A practical example illustrating our assumption of fixed initial hidden states is the medical scenario modeled by a factored-POMDP, where a patient's initial latent health factors could be confined to a narrow range and subsequently evolve over time, influenced by medical interventions and disease progression.
>
> In the revised manuscript, we have also incorporated the above clarification into the last paragraph of Section 3.3.
>
>
>  Comment 2.3:
> Generalization to More Complex Observation Models: Discuss how the approach could be extended to settings where observations are nonlinear transformations of hidden states. Providing insight into whether the model can handle more realistic assumptions would enhance its practical relevance.
>
>  Response 2.3:
>
> In a general POMDP, observations are typically (nonlinear) transformations of hidden states. Our factored-POMDP setting, however, requires that the observations are a subset of state factors. As noted in the third paragraph of the Discussion, this requirement imposes a limitation: converting a (high-dimensional) POMDP to a (low-dimensional) factored-POMDP often necessitates additional representation learning.
>
> To highlight this point earlier, we have revised the preamble to Subsection 2.2 to underscore why the factored-POMDP setting is crucial for constructing a causal graph of the transition model without edges within a time step (see Fig. 1). If the observation transformation mixes hidden factors, representation learning is needed to disentangle these factors—ensuring that, once conditioned on their parents, they remain independent.

---

> ### Author Response · Authors · 2025-03-30
> **Responses to the reviewer - Part 2**
>
> We thank the reviewer for the comments and have addressed them below. We are happy to respond to any further feedback.
>
>
>  Comment 2.4:
> Further Explanation of Additional Loss Terms: Expand on the reasoning behind Equations (8) and (9), describing each loss component in detail (Full NLL, Masked NLL, Causal NLL, Masked KL-Div, Causal KL-Div). Explain their individual purposes and how the differences between “Full” and “Masked” losses reveal the underlying causal structure.
>
>  Response 2.4:
>
> In the manuscript, we first derive the variational lower bound (VLB) of the log-likelihood for observation sequences in Eq. (22), under the assumption of a factorized state space, where the state vector comprises a concatenation of observable and hidden factors. Additionally, by assuming factorized transition dynamics, i.e., \(T(s_{t+1} | s_t, a_t) = \prod_{j=1}^{d_S} p(s_{t+1}^j | s_t, a_t)\), we factorize the hidden transition $p_{\theta_h}$, the observation transition $p_{\theta_o}$, and the hidden encoder $q_{\phi}$ in the VLB into products of distributions for individual factors (Eqs. (24–26)). This allows us to obtain Eq.(8), which we use to learn the factor-wise (full) transition terms $p_{\theta_{o}}(o_{t+1}^j | s_{t}, a_{t})$ and $p_{\theta_{h}}(h_{t+1}^j | s_{t}, a_{t})$ needed for evaluating the conditional mutual information (CMI) in Eqs. (12) and (13), respectively. These CMI terms are subsequently employed to estimate the causal edges between factors in the transition graph.
> Eq. (8) is further expanded into Eq. (9) by incorporating additional masked and causal losses, inspired by [1]. Specifically, the masked NLL and KL-divergence losses, calculated by randomly masking out one input state factor per sample in each mini-batch, serve as training regularizers and approximate the factor-wise masked transition terms $p_{\theta_{o}}(o_{t+1}^j | s_{t} \backslash s_{t}^i, a_{t})$ and $p_{\theta_{h}}(h_{t+1}^j | s_{t} \backslash s_{t}^i, a_{t})$, which are essential for computing the CMI as described in section 3.2. The causal NLL and KL-divergence losses are introduced to learn the causal transition dynamics $ p_{\theta_{o}}(o_{t+1}^j | PA_{o^j_{t+1}})$ and $p_{\theta_{h}}(h_{t+1}^j | PA_{h^j_{t+1}})$, where conditioning is performed solely on causal parents identified by the CMI-based inferred transition graph, rather than on the full set $\{s_t, a_t\}$.
>
> We have also incorporated the above explanation into the first three paragraphs of Subsection 3.4 in the revised manuscript.
>
> [1] Wang, Z., et al. (2022). Causal Dynamics Learning for Task-Independent State Abstraction.
>
>
>  Comment 2.5
> Define $\tau_t$ Explicitly: Provide a precise definition of $\tau_t$, clarifying whether it is an observable or hidden subset of the state. This information is crucial for understanding its role within the proposed framework.
>
>  Response 2.5:
>
> In our experiment, $\tau_t$ is defined as an episodic target state, whose components are observable and hidden factors. These components are randomly sampled at the beginning of each episode and remain fixed throughout the episode. The reward $r_t$ is defined as the number of matching factors between the current state $s_t=(o_t, h_t)=(o_{t}^1, \ldots, o_{t}^{d_O},h_{t}^1, \ldots, o_{t}^{d_H})$ and the target state $\tau_t=(\tau_t^1, \ldots, \tau_t^{d_O+d_H})$. Consequently, the supervised reward loss imposes additional constraints on the encoded hidden state $h_t$, preventing representational collapse.
> An illustrative example of the episodic target state $\tau_t$ is the desired health outcome of a patient, indicating recovery, which medical interventions aim to achieve. The reward can then be defined according to how closely the patient's current health state matches this target state.
>
> We have also incorporated the above explanation into the paragraph immediately following Eq. (12) in the revised manuscript.
>
>
>  Comment 2.6
> Broader Impact Concerns: Although the authors do not include a dedicated Broader Impact Statement in the paper, the discussion provided in Section 5 partially addresses the broader implications of their work. Specifically, they briefly touch on the limitation and considerations that arise when leveraging their method in real-world systems: multiple restrictive assumptions should be expanded to address the more complex challenges in the real world, like the independent observations and hidden states, the exploration and coverage of random policies, etc.
>
>  Response 2.6:
>
> In the Discussion, after mentioning all the limitations in the last two paragraphs, we have further added a penultimate sentence:
> "Our framework has limited applicability in the real world, since we require a factored-POMDP, fixed initial conditions for the hidden factors, restrictions on scalability to multiple hidden factors, and an offline dataset that covers all state-action transitions."

---

### Review · Reviewer_NnS5 · 2025-03-03

**Summary Of Contributions:**

# Summary

The authors assume a factored POMDP, where the Markovian state has some observed components and some unobserved components. They assume that the state transition is such that all the next state components are conditionally independents, given past state and past action. They also assume that the number of hidden state components $d_H$ is known.

For this setting, they derive a method that learns the causal conditonal independencies and the conditional probability distributions of the dynamics, using a variational inference method where the latent variables are the hidden state components, and where the observable are the observable state components. They first derive the ELBO as a lower bound on the likelihood of these observable variables, and then propose to rely on amortized variational inference to optimize this lower bound, using a variational approximation of the posterior in addition to the approximation of the likelihood. The first contribution is to identify a factorization of the posterior approximation, which appears to be new in the literature. This factorization first highlights that the posterior over the latent at time $t$ cannot be conditioned on the past only, which I think was already known in the litterature (Bayer et al., 2021; Becker et al., 2022), but interestingly this factorization of the posterior does not need past observations. Then, they extend an existing technique for learning causal dynamics (Wang et al., 2022) to POMDPs. In practice, this method requires that they jointly learn three instances of their variational model: one fully connected, one randomly masked, and one masked using an infered causal graph. This causal graph is infered using the estimated conditional mutual information between state components estimated from the first two models. The model results in a predictive model of the observations given the actions, which has internally infered a causal structure.

In the experiments, they furthermore assume that $d_H = 1$. They then show on two simple environments (but that have favorable properties for existing methods) that existing methods do not work as well as their method, for most considered metrics (graph accuracy and log likelihood).

**Audience:**

Yes

**Broader Impact Concerns:**

No concern.

**Claims And Evidence:**

Yes

**Requested Changes:**

# Requested Changes

- I would like you to either give the results comparing all methods on a bigger instance of the environment, or to better discuss why the method is currently limited and could not be scaled to a bigger instance (see my list of weaknesses for more details). Negative results should not be a criterion for rejection in my opinion.
- Better discussing the conditional independence assumption of the factored POMDP and clearly indicating whether it is a required assumption for the proposed factorization of the posterior approximation.

**Strengths And Weaknesses:**

# Strengths

- The paper reads very well, the explanations are clear, the mathematical formalization is good, and the mathematical derivations in appendices seem correct (and well detailed).
- There were many interesting discussions.
- The proposed factorization of the posterior is interesting, and motivates interesting future works.
- The research goal of discovering hidden variables and causal structures from interventional data is well motivated.

# Weaknesses

- The method was investigated on a single simple environment. It would have been interesting to at least compare all methods on a bigger instance of this environment: bigger state space and several hidden components. The fact that it could fail in such scenario is already well discussed in the paper, but I think that it would still be interesting to show these results. Moreover, I do not think that a negative result (where the proposed method does not work as good as another method) would be a criterion for rejection. If there are other reasons for not going further than a "proof of principle", I think it should be discussed.
- I have some questions that I think would deserve to be answered in the manuscript.
    - Does the factorization of the posterior approximation that you propose requires the conditional independence assumption of the state components in the transition distribution $T(s' | s, a)$? Whather the answer, I think that it would be interesting to mention it, even in the introduction.
    - What is a factor-wise masked MLP? Could you give a brief explanation in the appendix?
    - Could you better explain how the random factor $i$ that is masked is sampled? Is it a different $i$ for each sample of each minibatch?
- Some details are missing in the main text (only present in the conclusion/appendix).
    - The decomposition of $\theta$ in $\theta_o$ and $\theta_h$ is only given in the appendix.
    - The limitation of assuming an independent factorization of state variable is only stated in the conclusion, it would be good to explicitly give this assumption when introducing the factored POMDP.
- I think that it would be interesting to discuss the fact that the conditioning gap of history-based encoders is already known in the litterature (Bayer et al., 2021; Becker et al., 2022), for example in Remark 3. I think that the proposed fix of Becker et al. (2022) would corresponds to the "full trajectory encoder" with a variational RNN.

# Questions

- Are the parameters (for example $\theta_o$) shared for all three masked models? Or is there three sets of parameters?
- Is the extension to the inference over both hidden variables and noise variables straightforward? I guess that we loose the ability to uniquivocally identify the hidden variable in that case.

# Minor

- At the end of subsection 2.2, you explain what a factored POMDP is, starting from a factored MDP. It think that you should instead introduce the factor MDP as a special case of the factored POMDP, since the factored MDP was not introduced at this stage.
- At the beginning of section 3, you refer appendix A.3 instead of subsection 3.2.
- Equation (6) is not correct, you need to take the product over j and the product over i separately, which is not clear with that notation.
- In the end of subsection 3.1, you mention convergence of the DVAE to a trivial constant representation. Could you link this to the posterior collapse problem of VAEs? Is this related to the findings of Ni et al. (2024)?
- In subsection 3.3, maybe be it would be better to explain the modulo operation with $s_{t+1}^i$ instead of $s_{t+1}$, since it is applied separately on each component.
- In subsection 3.3, could you clarify whether your environment satisfies properties P1 and P2 in general (that is how I understood it), or whether these are additional constraints that you will enforce?
- I do not know what is a "categorical reward", and how we can define a "mean reward" from that? Is it a simple discretization of the continuous reward space, just for the sake of the prediction?
- In the discussions of the figure 4c in section 4, which are located in two different paragraphs, it is not so clear to me why it is a problem that $o^2_{t+1}$ is copied in the hidden state? I am sure that you know why you said that, but I was not able to understand.
- In the conclusion, I do not understand this sentence "when there is only one hidden factor, a single step of future information suffices to identify the hidden state up to a trivial transformation". Could it be better discussed in the main text?
- The PDF is really slow to load on my machine, maybe reducing the number of data point in the figures could help?

# References

- Bayer et al., 2021: "Mind the Gap When Conditioning Amortised Inference in Sequential Latent-Variable Models".
- Becker et al., 2022: "On Uncertainty in Deep State Space Models for Model-Based Reinforcement Learning".
- Wang et al., 2022: "Causal Dynamics Learning for Task-Independent State Abstraction".
- Ni et al., 2024: "Bridging State and History Representations: Undestanding Self-Predictive RL".

---

> ### Author Response · Authors · 2025-03-30
> **Responses to the reviewer - Part 1**
>
> We thank the reviewer for the comments and have addressed them below. We are happy to respond to any further feedback.
>
> Comment 1.1:
> The method was investigated on a single simple environment. It would have been interesting to at least compare all methods on a bigger instance of this environment: bigger state space and several hidden components. The fact that it could fail in such scenario is already well discussed in the paper, but I think that it would still be interesting to show these results. Moreover, I do not think that a negative result (where the proposed method does not work as good as another method) would be a criterion for rejection. If there are other reasons for not going further than a "proof of principle", I think it should be discussed.
>
> I would like you to either give the results comparing all methods on a bigger instance of the environment, or to better discuss why the method is currently limited and could not be scaled to a bigger instance (see my list of weaknesses for more details). Negative results should not be a criterion for rejection in my opinion.
>
> Response 1.1:
>
> We have compared all baseline methods regarding their scalability to a larger state space and two hidden factors using the modulo environment with a chain-structured transition graph. Although the losses for the hidden states and reward converge to sub-optimal values across all models—likely due to the inherent challenges of gradient-based optimization in discrete transition dynamics—our DVAE 1-step hindsight encoder consistently achieves superior training and evaluation performance compared to other baselines. These scalability analyses have been included in Appendices B.3 and B.4 of the revised manuscript, along with two sentences in the last para of Section 4. Please refer to these for detailed results.

---

> ### Author Response · Authors · 2025-03-30
> **Responses to the reviewer - Part 2**
>
> We thank the reviewer for the comments and have addressed them below. We are happy to respond to any further feedback.
>
> Comment 1.2:
> I have some questions that I think would deserve to be answered in the manuscript.
> 1. Does the factorization of the posterior approximation that you propose requires the conditional independence assumption of the state components in the transition distribution $T(s'|s,a)$? Whather the answer, I think that it would be interesting to mention it, even in the introduction.
> 2. What is a factor-wise masked MLP? Could you give a brief explanation in the appendix?
> 3. Could you better explain how the random factor $i$ that is masked is sampled? Is it a different $i$ for each sample of each minibatch?
>
> Better discussing the conditional independence assumption of the factored POMDP and clearly indicating whether it is a required assumption for the proposed factorization of the posterior approximation.
>
>  Response 1.2:
>
> 1. Technically speaking, the factorization of the posterior distribution derived in Eqs. (20–21) of the manuscript does not inherently require the assumption of factorized transition dynamics, i.e.,  $T(s_{t+1} | s_t, a_t) = \prod_{j=1}^{d_S} p(s_{t+1}^j | s_t, a_t)$ (as specified in the sixth point of the Factored-POMDP definition). It only necessitates that the state space itself be factorized, with each state vector represented as a concatenation of observed and hidden states (as described in the first four points of the Factored-POMDP definition). However, to explicitly learn factor-wise transitions and subsequently identify causal edges between factors, we utilize the additional assumption that the transition dynamics are also factorized, as illustrated in Eqs. (24–26). Both the factorized state space and the factorized transition dynamics assumptions are encapsulated within our definition of the Factored-POMDP.
>
> In Section 3.1 of the revised manuscript, we have improved the explanation of the factorization of both the posterior and the joint distributions. We added additional explanations and intuitive insights to clarify the mathematical formulas.
>
> 2. Each factor-wise masked MLP models the factorized transition probability to a state factor $s^j_{t+1}$ for $j$ in $1\dots d_S$. It accepts the complete set of current state factors and actions $\{s_t, a_t\}$ as inputs and outputs the distribution for a specific state factor $j$ at the next time step, $p(s_{t+1}^j | s_t, a_t)$. We employ  $d_S$ number of distinct masked MLPs, one for each state factor, to represent the factorized transition dynamics comprehensively. Moreover, the influence of a particular state or action factor on the output of the MLP can be selectively removed by masking the representation of that factor—setting it to negative infinity—and subsequently taking the maximum across the representations of all input factors within the MLP.
>
> We have also incorporated the above explanation into appendix A.5 of the revised manuscript.
>
> 3. For each sample in every mini-batch, the index $i$ of the masked factor used in calculating the masked losses is independently and identically sampled from a uniform distribution over $\{1, \ldots, d_S\}$.
>
> We have added the following clarification to the paragraph immediately following Eq. 11 in the revised manuscript:
>
> "For each factor $j$ in each sample within every mini-batch, the index $i$ is sampled independently and identically from a uniform distribution over $\{1, \ldots, d_S\}$."
>
>
>  Comment 1.3:
> Some details are missing in the main text (only present in the conclusion/appendix).
> 1. The decomposition of $\theta$ in $\theta_o$ and $\theta_h$ is only given in the appendix.
> 2. The limitation of assuming an independent factorization of state variable is only stated in the conclusion, it would be good to explicitly give this assumption when introducing the factored POMDP.
>
>  Response 1.3:
>
> 1. In the revised manuscript, we have decomposed $\theta$ into $\theta_o$ and $\theta_h$ in Eq. 2. We have also restructured Section 3 for greater clarity and intuition by providing a prelude with an intuitive overview of the derivations. Additionally, we split the original Section 3.1 into three subsections and reordered the material within that section and subsequent subsections.
> 2. In the first three paragraphs of Subsection 2.2, we have enhanced the explanation of the assumptions required for a factored-POMDP and clarified how it differs from a general POMDP.
> Please let us know if the revised manuscript addresses your points and please suggest any additional changes.

---

> ### Author Response · Authors · 2025-03-30
> **Responses to the reviewer - Part 3**
>
> We thank the reviewer for the comments and have addressed them below. We are happy to respond to any further feedback.
>
>  Comment 1.4:
> I think that it would be interesting to discuss the fact that the conditioning gap of history-based encoders is already known in the literature (Bayer et al., 2021; Becker et al., 2022), for example in Remark 3. I think that the proposed fix of Becker et al. (2022) would corresponds to the "full trajectory encoder" with a variational RNN.
>
>  Response 1.4:
>
> References [1, 2] define the conditioning gap as a specific inference error that arises when the variational posterior is under-conditioned. While the true posterior distribution depends on the entire sequence of observations, approximate posteriors typically depend only on past observations (e.g., using history-based encoders). Omitting parts of the conditioning information forces the model to merge several distinct true posterior distributions into a single compromised distribution, ultimately resulting in a suboptimal generative model.
>
> A potential solution to the conditioning gap is provided by the full-trajectory-based encoder proposed in [2]. This encoder employs closed-form Kalman smoothing (forward-backward filtering) under linear–Gaussian assumptions of latent dynamics.
>
> We have mentioned the known issue of the conditioning gap associated with history-based encoders, along with relevant citations, in Remark 3 of the revised manuscript.
>
> [1] Bayer et al., (2021). Mind the Gap When Conditioning Amortised Inference in Sequential Latent-Variable Models.
>
> [2] Becker et al., (2022). On Uncertainty in Deep State Space Models for Model-Based Reinforcement Learning.
>
>
>  Comment 1.5:
> Are the parameters (for example $\theta_o$) shared for all three masked models? Or is there three sets of parameters?
>
>  Response 1.5:
>
> The network parameters $\theta_o, \theta_h$ and $\phi$ ($\overline{\phi}$) are shared across all three types of losses. Three distinct masks (no mask, random mask, and causal mask) are applied to the same input $\{s_t, a_t\}$, producing different distributions that are utilized separately for each of the three loss computations.
>
> We have also added a clarifying sentence to the paragraph immediately following Eq. 11 in the revised manuscript.
>
>
>  Comment 1.6:
> Is the extension to the inference over both hidden variables and noise variables straightforward? I guess that we loose the ability to unequivocally identify the hidden variable in that case.
>
>  Response 1.6:
>
> In the revised manuscript, we have incorporated the following proposal to extend our framework to infer noise and additional hidden factors in Appendix B.5.
>
> In the noisy observation setting with a single hidden factor $h_t$, we can introduce $d_O$ additional noise encoders $q_{\phi_\epsilon}(\epsilon_t^i | o_t, h_t,  a_t)$, one for each noise variable $\epsilon_t^i$. Each noise encoder is conditioned on the latent state $h_t$, inferred by our DVAE 1-step hindsight encoder $q_{\phi}(h_{t} | h_{t-1}, o_{t-1: t+1}, a_{t-1: t+1})$, as well as on the current observation $o_t$ and action $a_t$.
> To ensure exogeneity of the noise factors $\epsilon_t^1, \dots, \epsilon_t^{d_O}$, we require each of them to be independent of the state-action pair $o_t, h_t, a_t$. Analogous to the CMI for the next hidden state in Eq.  (11), we measure the factor-wise mutual information between each noise factor $\epsilon_t^i$ and the state-action pair $o_t, h_t, a_t $ as
> \begin{equation}
> I_{\phi_\epsilon}(\epsilon_t^i; o_t, h_t,  a_t) = E_{o_t, h_t, a_t \sim \mathcal{D}, q_{\phi}}[D_{\mathrm {KL}}(q_{\phi_\epsilon}(\epsilon_t^i | o_t, h_t,  a_t) \parallel q_{\phi_\epsilon}(\epsilon_t^i))]
> \end{equation}
>
> By driving the sum of these factor-wise mutual information terms across all noise factors to zero,
> \begin{equation}
> \min_{\phi_\epsilon} \sum_{i=1}^{d_O} I_{\phi_\epsilon}(\epsilon_t^i; o_t, h_t,  a_t)
> \end{equation}
> we enforce independence between the noise factors and the state-action pair, thereby preserving the exogeneity of the noise.
>
> However, if multiple current hidden factors collide on a conditioned future observation factor in the unrolled transition graph (see the last two sentences in the paragraph immediately following Eq. 3 for details), the path between these factors becomes unblocked. In this situation, the DVAE-based encoder can no longer unequivocally disentangle these hidden factors, let alone separate them from the noise.

---

> ### Author Response · Authors · 2025-03-30
> **Responses to the reviewer - Part 4**
>
> We thank the reviewer for the comments and have addressed them below. We are happy to respond to any further feedback.
>
> Minor:
>
>  Comment 1.7:
> At the end of subsection 2.2, you explain what a factored POMDP is, starting from a factored MDP. I think that you should instead introduce the factor MDP as a special case of the factored POMDP, since the factored MDP was not introduced at this stage.
>
>  Response 1.7:
>
> Indeed, the reviewer is correct. In fact, Subsection 2.2 is entirely general for any factored-POMDP (with a factored-MDP as a special case). Therefore, in the revised manuscript, we have rephrased the line above Eq. (1) to refer to a factored-POMDP instead of a factored-MDP, and we have removed the statement about constructing a factored-POMDP from a factored-MDP by hiding some factors (the first line in the last paragraph of Subsection 2.2).
>
>
>  Comment 1.8:
> Equation (6) is not correct, you need to take the product over j and the product over i separately, which is not clear with that notation.
>
>  Response 1.8:
>
> We thank the reviewer for highlighting this issue. To avoid potential confusion, we have rewritten Eq. (2) explicitly by separating the products over indices $i$ and $j$ as follows:
> \begin{align}
> p(o_{1:T}, h_{1:T} | a_{1:T}) =& \prod_{t=0}^{T-1}\prod_{j=1}^{d_H} p(h_{t+1}^j | s_t, a_t) \prod_{i=1}^{d_O} p(o_{t+1}^i | s_t, a_t)
> \end{align}
>
>
>  Comment 1.9:
> In the end of subsection 3.1, you mention convergence of the DVAE to a trivial constant representation. Could you link this to the posterior collapse problem of VAEs? Is this related to the findings of Ni et al. (2024)?
>
>  Response 1.9:
>
> The phenomenon of posterior collapse in VAEs described in [1] and the collapse of self-predictive representations in RL discussed in [2] are closely related but framed within different contexts.
> Posterior collapse occurs in VAEs when the variational posterior $q_{\phi}(z | x)$ fails to learn a meaningful latent representation from the observations, collapsing instead to match the prior distribution  $p(z)$. Mathematically, this is expressed as $q_{\phi}(z | x)=p(z)$ for all $x$.
> In self-predictive RL, representation collapse arises when the latent state $q_{\phi}(z | h)$, encoded based on the history $h$, and the predicted next latent state $p_{\theta}(z' | z, a)$ collapse to a constant representation $c$, i.e., $p_{\theta}(z' | z, a)=q_{\phi}(z' | h')=c$. This issue emerges due to the bootstrapped nature of latent representations used in self-predictive learning.
>
> In the revised manuscript, we have added a summary of the above clarification in the fourth paragraph of Section 3.4.
>
> [1] He, J., et al. (2019). Lagging inference networks and posterior collapse in variational autoencoders.
>
> [2] Ni, T., et al. (2024). Bridging state and history representations: Understanding self-predictive rl.
>
>
>  Comment 1.10:
> In subsection 3.3, maybe it would be better to explain the modulo operation with $s^i_{t+1}$ instead of $s_{t+1}$, since it is applied separately on each component.
>
>  Response 1.10:
>
> In Subection 3.5 of the revised manuscript, we succinctly define the modulo transition dynamics in a vectorized form as $s_{t+1} := ( A s_t + a_t + \epsilon_{t+1} )\ \text{mod}\ l$. More specifically, the modulo operation is applied element-wise to the $d_S$-dimensional vector $A s_t + a_t + \epsilon_{t+1}$, yielding the next state vector $s_{t+1}$, which has the same dimensionality. In Section 4, we further represent the transition dynamics explicitly for each individual factor within a particular environmental setup.
>
>
>  Comment 1.11:
> In subsection 3.3, could you clarify whether your environment satisfies properties P1 and P2 in general (that is how I understood it), or whether these are additional constraints that you will enforce?
>
>  Response 1.11:
>
> Our modulo environment generally satisfies both properties. For property P1, various transition graph structures are permissible, provided that at least one child factor at time step  $t+1$ of the hidden factor at time step $t$ is observable. For property P2, given that the state and action spaces are discrete and the transition dynamics involve an element-wise modulo operation applied to a linear combination of the current state-action pair and noise—specifically, $s_{t+1} := ( A s_t + a_t + \epsilon_{t+1} )\ \text{mod}\ l$—the mapping from $s_t$ to $s_{t+1}$ is always bijective for any fixed $a_t$ and $\epsilon_t$. Additionally, we assume that the initial hidden state remains fixed across episodes. For further details regarding state initialization, please refer to our response 2.2 to Reviewer uoD6.
>
> In the revised manuscript, we have also incorporated the above clarification into the second paragraph of Section 3.5.

---

> ### Author Response · Authors · 2025-03-30
> **Responses to the reviewer - Part 5**
>
> We thank the reviewer for the comments and have addressed them below. We are happy to respond to any further feedback.
>
> Minor:
>
>  Comment 1.12:
> I do not know what is a "categorical reward", and how we can define a "mean reward" from that? Is it a simple discretization of the continuous reward space, just for the sake of the prediction?
>
>  Response 1.12:
>
> A categorical reward refers to a reward that takes discrete values. Our reward predictor outputs a categorical distribution (a probability distribution over discrete outcomes), and we define the mean reward as the expectation (mean) of this distribution.
>
> We have added a clarifying sentence at the end of the paragraph immediately following Eq. 12 in the revised manuscript.
>
>
>  Comment 1.13:
> In the discussions of the figure 4c in section 4, which are located in two different paragraphs, it is not so clear to me why it is a problem that $o^2_{t+1}$ is copied in the hidden state? I am sure that you know why you said that, but I was not able to understand.
>
>  Response 1.13:
>
> In this scenario, if the encoder infers the hidden state $h_{t}^1$ as a direct copy of the observable $o^2_{t+1}$, the transition model $o_{t+1}^2 = f_2(h_{t}^1, o_{t}^2, a^3_t, \epsilon_{t}^3 )$ would degenerate into a trivial identity mapping $o_{t+1}^2 = h_{t}^1$. This pathological solution effectively disregards the other conditioning factors $\{ o_{t}^2, a^3_t, \epsilon_{t}^3 \}$ and, consequently, results in incorrect edges within the estimated transition graph.
>
> We have incorporated the above explanation into the second paragraph following Table 1 in the revised manuscript.
>
>
>  Comment 1.14:
> In the conclusion, I do not understand this sentence "when there is only one hidden factor, a single step of future information suffices to identify the hidden state up to a trivial transformation". Could it be better discussed in the main text?
>
>  Response 1.14:
>
> In the case of a single hidden factor, we require only one-step future information to identify the hidden state. The discrete value of the inferred hidden state can then be mapped to that of the true hidden state through an invertible linear transformation. To avoid confusion, we have replaced the sentences above with the original sentence in the revised manuscript. For further discussion on the identifiability of current hidden states in the VAE setting, as well as empirical validation of this transformation, please refer to Remark 2 and Table 1 in the manuscript.

---

> ### Comment · Reviewer_NnS5 · 2025-04-14
>
> Dear Authors,
>
> I would like to thank you for your hard work on the revised manuscript. In particular, I thank you for clarifying the conditional independence assumption, and for having added experiments on larger problems. I think that these are further motivating your approach, and I personally find them interesting. I have carefully read your rebuttals and the relevants parts of the revised manuscript. All my comments and questions have been adressed and answered.
>
> One minor point remaining is the product on hidden and observable factors, which is still incorrect. Indeed, note that,
> $$
>     \left( \prod_{i=1}^{I} x_i \right) \left( \prod_{j=1}^{J} y_j \right)
>     \neq
>     \prod_{i=1}^{I} x_i \prod_{j=1}^{J} y_j
>     =
>     \prod_{i=1}^{I} \prod_{j=1}^{J} x_i y_j.
> $$

---

> > ### Author Response · Authors · 2025-05-01
> >
> > Dear Reviewer NnS5,
> >
> > Thanks a lot for appreciating our work and going through in detail and confirming that all your comments and questions have been addressed.
> >
> > About the minor remaining point, you are indeed correct about the placement of parentheses in Eqn. 2 of our manuscript. This has been rectified as below:
> > $$ p(o_{1:T}, h_{1:T} | a_{1:T}) = \prod_{t=0}^{T-1} \left(\prod_{i=1}^{d_O} p(o_{t+1}^i | s_t, a_t)\right) \left(\prod_{j=1}^{d_H} p(h_{t+1}^j | s_t, a_t)\right) $$
> > The updated pdf has the intermediate steps as well.
> >
> > Thanks,
> > Authors.

---

### Review · Reviewer_8EuP · 2025-03-08

**Summary Of Contributions:**

This paper proposes a novel encoder for learning latent states and transition dynamics in a Partially Observable Markov Decision Process (POMDP) using offline trajectories. The decoder follows the framework of Dynamical Variational Auto-Encoders (DVAE) and is based on variational inference and the evidence lower bound. The key contribution of the proposed encoder is leveraging future observations and actions as hindsight information to predict the current hidden variables. The paper focuses on the factored-POMDP setting and conducts experiments in a synthetic environment. Empirical results demonstrate that the proposed encoder outperforms baseline hidden state encoders.

**Audience:**

Yes

**Claims And Evidence:**

No

**Requested Changes:**

a. I recommend conducting experiments in real-world environments or highly non-trivial synthetic settings where predicting latent variables is challenging. Such experimental results would better demonstrate the effectiveness of the proposed encoder.

b. The writing in Section 3 should be improved by adding more explanations and intuitions for the mathematical formulas. The current version is math-heavy and difficult to follow.

Minor Comments:
In the last third line of Page 3, change subsection A.3 → subsection 3.2.

In the third line below Eq. 7, the term $q_{\phi}(h_{t+1}^j \mid h_t,o_t,a_t)$ should be $q_{\phi}(h_{t+1}^j \mid h_t,o_{t:T},a_{t:T})$.

Some relevant related works also employ future-dependent functions in the offline POMDP setting:

[1] Uehara, Masatoshi, et al. "Future-dependent value-based off-policy evaluation in POMDPs." Advances in Neural Information Processing Systems, 36 (2023): 15991-16008.

[2] Zhang, Yuheng, and Nan Jiang. "On the curses of future and history in future-dependent value functions for off-policy evaluation." arXiv preprint, arXiv:2402.14703 (2024).

**Strengths And Weaknesses:**

Strengths:

This paper addresses a fundamental problem in POMDPs—learning hidden variables and transition dynamics. To tackle this, it proposes a novel encoder that leverages hindsight information, incorporating future observations and actions. The approach is well-motivated and conceptually sound.

Weaknesses:

My primary concern is the persuasiveness of the empirical results. The authors only conduct experiments on a synthetic 1-hidden factored-POMDP, which may not sufficiently demonstrate the encoder's effectiveness in more complex scenarios. As shown in the figures in Section 4, multiple baseline encoders perform similarly to the proposed encoder. Additionally, the tables indicate that several encoders achieve high accuracy in predicting $h^1$, with the proposed encoder even reaching 100% accuracy on most metrics. This raises concerns about whether the synthetic environment is too simplistic and whether the proposed method would remain effective in more complex, real-world settings.

The writing of this paper also needs improvement, particularly in Section 3. The authors do not provide sufficient intuition or interpretations for the mathematical formulas, making it difficult for readers to follow.

---

> ### Author Response · Authors · 2025-03-30
> **Responses to the reviewer - Part 1**
>
> We thank the reviewer for the comments and have addressed them below. We are happy to respond to any further feedback.
>
>  Comment 3.1:
> My primary concern is the persuasiveness of the empirical results. The authors only conduct experiments on a synthetic 1-hidden factored-POMDP, which may not sufficiently demonstrate the encoder's effectiveness in more complex scenarios. As shown in the figures in Section 4, multiple baseline encoders perform similarly to the proposed encoder. Additionally, the tables indicate that several encoders achieve high accuracy in predicting $h^1$, with the proposed encoder even reaching 100\% accuracy on most metrics. This raises concerns about whether the synthetic environment is too simplistic and whether the proposed method would remain effective in more complex, real-world settings.
>
> I recommend conducting experiments in real-world environments or highly non-trivial synthetic settings where predicting latent variables is challenging. Such experimental results would better demonstrate the effectiveness of the proposed encoder.
>
>  Response 3.1:
>
> Our contributions in this paper are to show:
> (1) the conditions of a factored-POMDP under which the transition function can be considered as a structural causal model (sec. 2.2);
> (2) the necessity of having an encoder $q(h_{t+1}^j|h_t,o_{t:T},a_{t:T})$ for the hidden factor $h_{t+1}^j$ that is conditioned on the 1-step past full state-action, current and future observations and actions (sec. 3.1);
> (3) the application of a DVAE framework to learn the encoder and transition model (sec. 3.2);
> (4)  the estimation of edges in the causal graph of the transition model (sec. 3.3 and 3.4) ;
> (5) a proof-of-principle demonstration for a 1-hidden factored-POMDP (sec. 3.5 and sec. 4).
>
> We have added new simulations to demonstrate further scalability to larger state spaces ($d_S=5, l=5$) and two hidden factors in Appendices B.3 and B.4 respectively, using the modulo environment with a chain-structured transition graph. Our DVAE 1-step hindsight encoder consistently outperforms baseline methods in both training and evaluation, although the hidden factors and transition graph are not learned perfectly. Also, losses for hidden states and reward converge to sub-optimal values across all models, likely due to the inherent challenges of gradient-based optimization in discrete transition dynamics.
>
> In the revised manuscript, we have added lines in the last paragraph of Sec. 4 outlining this scalability analysis, with references to Appendix B.3 and B.4. for details.
>
> Further, we acknowledge several limitations of the current work:
> (1) We do not tackle the extremely important and challenging problem of representation learning to extract independent state factors to convert a real-world (high-dimensional) POMDP into a (low-dimensional) factored-POMDP, as needed for our framework (Discussion para 3, we also added new text in Sec. 2.2).
> (2) While our framework achieves consistently better training and evaluation performance than the others on scaling,  it does not learn the hidden state and the transition graph with 100\% accuracy, as shown in the new simulation results, detailed above.
> (3) We may not be able to disentangle multiple hidden factors, that collide on an observable factor in the transition graph (Appendix B.5)
>
> In the Discussion, after mentioning all the limitations in the last two paragraphs, we have further added a penultimate sentence: "Our framework has limited applicability in the real world, since we require a factored-POMDP, fixed initial conditions for the hidden factors, restrictions on scalability to multiple hidden factors, and an offline dataset that covers all state-action transitions. Addressing these areas would support further scaling and generalization of the framework."
>
> We believe that our current claims are justified by our experiments. Further experiments on real-world environments are beyond the scope of the paper, as these would require representation learning, further theory, and additional time and resources unavailable to us presently.

---

> ### Author Response · Authors · 2025-03-30
> **Responses to the reviewer - Part 2**
>
> We thank the reviewer for the comments and have addressed them below. We are happy to respond to any further feedback.
>
>  Comment 3.2:
> The writing of this paper also needs improvement, particularly in Section 3. The authors do not provide sufficient intuition or interpretations for the mathematical formulas, making it difficult for readers to follow.
>
> The writing in Section 3 should be improved by adding more explanations and intuitions for the mathematical formulas. The current version is math-heavy and difficult to follow.
>
>  Response 3.2:
>
> We have restructured Section 3 completely, with a prelude providing an intuitive overview of the derivations in the subsections. The decomposition into time and state factors (earlier in Appendix) is now incorporated in the main text (subsection 3.1), before introducing the VAE (subsection 3.2), to highlight that the dependence of the current hidden factors on 1-step past, current and full future, is valid without the VAE approximation. We have also split up the earlier section 3.1 into 3 subsections and reordered the material therein and in the subsequent subsections. In particular, the CMI is explained first in a subsection, before extra loss terms to compute CMI are added to the VAE loss. The loss terms are also explained in more detail, in particular how the masking of factors is implemented to compute the masked loss terms.
> Please let us know if the revised manuscript addresses your concerns, and please suggest any additional changes.
>
> Minor:
>
>  Comment 3.3:
> In the last third line of Page 3, change subsection A.3 → subsection 3.2.
>
> In the third line below Eq. 7, the term $q_\phi(h^j_{t+1} | h_t, o_t, a_t)$ should be $q_\phi(h^j_{t+1} | h_t, o_{t:T}, a_{t:T})$.
>
> Some relevant related works also employ future-dependent functions in the offline POMDP setting:
> [1] Uehara, Masatoshi, et al. "Future-dependent value-based off-policy evaluation in POMDPs." Advances in Neural Information Processing Systems, 36 (2023): 15991-16008.
> [2] Zhang, Yuheng, and Nan Jiang. "On the curses of future and history in future-dependent value functions for off-policy evaluation." arXiv preprint, arXiv:2402.14703 (2024).
>
>  Response 3.3:
>
> We thank the reviewer for pointing these out.
> In the revised manuscript, we have corrected the typos and added sentences in the first paragraph of Sec. 5 to summarize these related works.

---

### Decision · Action_Editor_dki1 · 2025-05-01

**Recommendation:** Reject

**Comment:**

The paper received mixed reviews, with two [8EuP,uoD6] leaning toward rejection and one [NnS5] leaning towards acceptance.

The reviewers appreciated several aspects of the work:
+ It was appreciated that the method addressed a task of learning POMDP hidden variables and dynamics, which was considered well motivated [Nns5] and a fundamental POMDP task [8EuP].
+ The proposed encoder was considered novel [8EuP].
+ The posterior factorization was considered interesting [NnS5]
+ The approach was considered well-motivated and sound [8EuP].
+ The paper was considered clearly written [uoD6,NnS5] and readable [NnS5], including good mathematical formalization and derivations [NnS5]
+ The discussion was considered interesting [NnS5]

However, several concerns were raised in the reviews.
- Better explanation or demonstration of the benefit of using future information was requested [uoD6]. Similarly, clarifying discussion on identifiability of hidden states without future information was requested [uoD6]
- Having experiments only on a synthetic POMDP was criticized as not sufficient [8EuP,NnS5], and experiments were desired showing scaling to bigger environments [NnS5] and in real-world (or non-trivial synthetic) settings [8EuP]. Similarly, discussion of extension of the approach to more complex observation models was requested [uoD6]
- The synthetic entironment was criticized as too simplistic, with several encoders achieving high accuracy and several baseline encoders were seen as performing similarly to the proposed one [8EuP].
- There was a clarity concern about how data is collected under the policy assumption [uoD6]
- Clarification of the assumptions on the initial hidden state and initial observation was requested [uoD6]
- Clarification of the rationale of the multiple loss components was requested [uoD6]
- Several technical clarifications were requested [NnS5,uoD6]
- Writing improvements for improved intuition were desired [8EuP]
- Some additional discussion of results from previous literature was desired [NnS5]

Authors provided responses addressing some of these concerns, where authors stated e.g. that they had added new simulations to demonstrate scalability and performed some rewriting, as well as clarifying several technical questions. After the author responses and discussion, the author rebuttal was appreciated by one reviewer [NnS5]. However, concerns remained regarding the use of only synthetic experiments [8EuP] and practical applicability in real-world scenarios due to strong assumptions [uoD6].

Overall, the remaining concerns including the synthetic experiments and practical applicability seem sufficiently strong that the paper does not seem ready to be presented in TMLR.

**Audience:**

The topic of the paper relates to the general task of learning POMDP hidden variables and dynamics, here in a factored-POMDP setting, which seems relevant to part of the audience interested in POMDP modeling.

**Claims And Evidence:**

Some of the claims and mathematical formulation were considered clear but there are concerns about convincingness of the evidence, in particular the use of only synthetic experiments and practical applicability in real-world scenarios (see the "Comment" field for details).